# INSIHGT: an accessible multi-scale, multi-modal 3D spatial biology platform

Chun Ngo Yau [1,2,8], Jacky Tin Shing Hung [1,2,8], Robert A. A. Campbell [3], Thomas Chun Yip Wong[1,2], Bei Huang [1,2,4], Ben Tin Yan Wong [1,2], Nick King Ngai Chow[1,2], Lichun Zhang[1,2], Eldric Pui Lam Tsoi[1,2], Yuqi Tan [5], Joshua Jing Xi Li [6], Yun Kwok Wing [1,2,4] & Hei Ming Lai [1,2,7] ✉

Biological systems are complex, encompassing intertwined spatial, molecular and functional features. However, methodological constraints limit the completeness of information that can be extracted. Here, we report the development of INSIHGT, a non-destructive, accessible three-dimensional (3D) spatial biology method utilizing superchaotropes and host-guest chemistry to achieve homogeneous, deep penetration of macromolecular probes up to centimeter scales, providing reliable semi-quantitative signals throughout the tissue volume. Diverse antigens, mRNAs, neurotransmitters, and post-translational modifications are well-preserved and simultaneously visualized. INSIHGT also allows multi-round, highly multiplexed 3D molecular probing and is compatible with downstream traditional histology and nucleic acid sequencing. With INSIHGT, we map undescribed podocyte-to-parietal epithelial cell microfilaments in mouse glomeruli and neurofilament-intensive inclusion bodies in the human cerebellum, and identify NPY-proximal cell types defined by spatial morpho-proteomics in mouse hypothalamus. We anticipate that INSIHGT can form the foundations for 3D spatial multi-omics technology development and holistic systems biology studies.

The complexity of biological systems mandates high-dimensional measurements to obtain an integrative understanding. However, measurements are inevitably perturbative, affecting the authenticity of the retrieved information. Spatially resolved transcriptomics[1] and highly multiplexed immunohistochemistry (IHC)[2] have proven to be powerful approaches to extract spatial molecular insights from tissue slices, but the two-dimensional (2D) readout limits the representativeness of the information extracted. Meanwhile, three-dimensional (3D) multiplexed visualization of tissue structural and molecular features can reveal previously unknown organization principles[3,4]. Optical

tissue clearing technologies promises to reveal the authentic 3D nature of tissue architecture and molecular distributions[5]. Despite its significant advancements, the achievable depths of probe penetration limits the depth of analysis[6]. The limited penetration of antibodies in 3D IHC represents one of the most significant barrier to 3D spatial biology[6].

In recent years, multiple creative solutions have been proposed for deep immunohistochemistry[7–10]. However, an accessible technology that balances the authenticity and volume of data extracted is still lacking. For example, signal homogeneity across penetration depth is

[1]Department of Psychiatry, Faculty of Medicine, The Chinese University of Hong Kong, Hong Kong SAR, China. [2]Li Ka Shing Institute of Health Sciences, Faculty of Medicine, The Chinese University of Hong Kong, Hong Kong SAR, China. [3]Sainsbury Wellcome Centre for Neural Circuits and Behaviour, University College London, London, UK. [4]Li Chiu Kong Family Sleep Assessment Unit, Department of Psychiatry, The Chinese University of Hong Kong, Hong Kong SAR, China. [5]Department of Microbiology and Immunology, Stanford University, Stanford, CA, USA. [6]Department of Pathology, School of Clinical Medicine, The University of Hong Kong, Queen Mary Hospital, Hong Kong SAR, China. [7]Department of Chemical Pathology, Faculty of Medicine, The Chinese University of Hong Kong, Hong Kong SAR, China. [8]These authors contributed equally: Chun Ngo Yau, Jacky Tin Shing Hung. ✉e-mail: hmlai@cuhk.edu.hk

suboptimal with most methods, where probes preferentially deposit near the tissue surface and complicates downstream quantitative protein expression determination[6,11]. The homogeneous penetration can only be attained either through complicated operations or equipment[11–13], or extensive tissue permeabilization[14,15] or incubation times measuring in weeks[8]. These shortcomings hinder the wide adoption of 3D tissue analysis in research and renders them unsatisfactory for clinical translation.

Here, we report the development of In situ Host-Guest Chemistry for Three-dimensional Histology (INSIHGT). INSIHGT is a user-friendly 3D histochemistry method, featuring (1) homogeneous probe penetration up to centimeter depths, (2) producing quantitative, highly specific immunostaining signals, (3) a fast and affordable workflow to accommodate different tissue sizes and shapes, (4) simple immersion-based staining at room temperature, thus easily adopted in any laboratory and ready for scaling and automate, and (5) uses off-the-shelf antibodies or probes and is directly applicable to otherwise unlabeled mouse and human tissues fixed with paraformaldehyde only. INSIHGT was developed based on the manipulation of macromolelular diffusiophoresis using *closo*-dodecahydrododecaborate $[B_{12}H_{12}]^{2-16}$ and a γ-cyclodextrin derivative. If tissue clearing is required, INSIHGT works best with solvent-based clearing methods[17–19].

## Results
### Modulation of antibody-antigen binding for enhanced probe penetration

The limited penetration of macromolecular probes in complex biological systems belongs to the broader subject of transport phenomena, where diffusion and advections respectively drive the dissipation and directional drift of mass, energy and momentum. When biomolecules such as proteins are involved, the (bio)molecular fluxes are additionally determined by binding reactions, which can significantly deplete biomolecules due to their high binding affinities and low concentrations employed - a "reaction barrier" to deep antibody penetration. This is first described and postulated by Renier et al.[17] (as in immunolabeling-enabled three-dimensional imaging of solvent-cleared organs, iDISCO) and Murray et al.[14] (as in system-wide control of interaction time and kinetics of chemicals, SWITCH), and the latter further showed that the modulation of antibody-antigen (Ab-Ag) binding affinity (SWITCH labeling) can lead to homogeneous penetration of up to 1 mm for an anti-Histone H3 antibody using low concentrations of sodium dodecyl sulfate (SDS). Other techniques similarly utilizes urea[8], sodium deoxycholate[12], and heat[9] to modulate antibody-antigen binding.

However, others and we observed a general compromise between antibody labelling quality, penetration depth and uniformity, and duration of incubation. Deep penetration invariably requires long incubation times with inhomogeneous signal across depth, while faster methods leads to weak or nonspecific staining, as well as non-uniform penetration[8,9,17]. Specifically, the use of SDS for deep labelling with SWITCH labelling has only been demonstrated for a handful of antigens (e.g., Histone H3[14], NeuN[20], ColIV, αSMA, and TubIII[21]). It was found that deep staining with SDS was not universally applicable[20], resulting in weak calbindin staining[22], insufficient staining depth for β-amyloid plaques[23], and often required tailored refinement of buffer concentration[24]. In our validation data, we similarly observed the variable performance when SDS is co-applied with antibodies (Supplementary Fig. 1). Furthermore, although adding antibodies or probes theoretically improves penetration via steep concentration gradients, either the cost becomes prohibitive or it produces a biased representation of rimmed surface staining pattern[6,8], especially for densely expressed binding targets. In the most extreme cases, the superficial staining signal would saturate microscope detectors while the core remains unstained (Supplementary Fig. 2).

Nonetheless, the conception of modulating antibody-antigen binding kinetics as a means to control probe flux through tissues is highly attractive[12,14], given the simplicity, scalability, and affordability should the method be robust and generalizable. We postulated that the reason for the highly variable performance of SDS-assisted deep immunostaining is two-fold: the denaturation of antibodies beyond reparability, and the ineffective reinstatement of binding reactions. This prompted us to search for alternative approaches that can tune biomolecular binding affinities while preserving both macromolecular probe mobility and stability. In addition, the negation of the modulatory effect should be efficient and robust to reinstate biomolecular reactions within the complex tissue environment. Therefore, here we aim to develop a fast, equipment-free, deep and uniform multiplexed immunostaining method, which will help bring 3D histology to any basic laboratories.

### Boron cluster host–guest chemistry for in situ macromolecular probe mobility control

Our initial attempts by using heat and the GroEL-GroES system to denature and refold antibodies in situ respectively have proved unsuccessful (Supplementary Fig. 1). We thus switched from the natural molecular chaperones to artificial ones using milder detergents (e.g., sodium deoxycholate (SDC) and 3-([3-Cholamidopropyl]dimethylammonio)- 2-hydroxy-1-propanesulfonate i.e., CHAPSO) and their charge-complementary, size-matched host-complexing agents (e.g., β-cyclodextrins and their derivatives such as heptakis-(6-amino-6-deoxy)-beta-cyclodextrin, i.e., 6NβCD), which improved antibody penetration and staining success rate (Supplementary Fig. 3). However, despite extensive optimization on the structure and derivatization on the detergents and their size- and charge-complementary cyclodextrins, they still have limited generality for a panel of antibodies tested (Supplementary Fig. 3), producing nonspecific vascular precipitates or nuclear stainings. We then explored the use of chaotropes, which are known to solubilize proteins with enhanced antibody penetration[8]. However, these approaches require long incubation times with extensive tissue pre-processing. Furthermore, higher concentrations of chaotropes often denature proteins as they directly interact with various protein residues and backbone[25,26] (Fig. 1a, b).

We hence focus on testing weakly coordinating superchaotropes (WCS), a class of chemicals that we hypothesized to inhibit antibody-antigen interactions while preserving their structure and hence functions (Fig. 1a, b). We searched for weakly coordinating ions based on their utility in isolating extremely electrophilic species for X-ray crystallography, or as conjugate bases of superacids. We can then select a subset of these coordinatively inert ionic species that possess high chaotropicity as candidates for our deep immunostaining purpose. After antibodies and WCS have been homogeneously distributed throughout the tissue matrix, measures must be taken to negate the superchaotropicity to reinstate inter-biomolecular interactions in a bio-orthogonal and system-wide manner. To do so, we took advantage of the enthalpy-driven chaotropic assembly reaction, where the activities of superchaotropes can be effectively negated with supramolecular hosts in situ, reactivating interactions between the macromolecular probes and their tissue targets.

Based on the above analysis, we designed a scalable deep molecular phenotyping method, performed in two stages: a first infiltrative stage where macromolecular probes co-diffuse homogeneously with WCS with minimized reaction barriers, followed by the addition of macrocyclic compounds for in situ host-guest reactions to reinstate antibody-antigen binding. With a much-narrowed list of chemicals to screen, we first benchmarked the performances of several putative WCS host-guest systems using a standard protocol as previously

published[6,8,9] (Supplementary Fig. 4). These include perrhenate/α-cyclodextrin (ReO$_4^-$/αCD), ferrocenium/βCD ([Fe(C$_5$H$_5$)$_2$]$^+$/βCD), *closo*-dodecaborate ions ([B$_{12}$X$_{12}$]$^{2-}$/γCD (where X = H, Cl, Br, or I)), metallacarborane ([Co(7,8-C$_2$B$_9$H$_{11}$)$_2$]$^-$/γCD), and polyoxometalates ([PM$_{12}$O$_{40}$]$^{3-}$/γCD (where M = Mo, or W)) (Fig. 1c, d). Group 5 and 6 halide clusters and rhenium chalcogenide clusters such as [Ta$_6$Br$_{12}$]$^{2+}$, [Mo$_6$Cl$_{14}$]$^{2-}$ and {Re$_6$Se$_8$}$^{2+}$ derivatives were excluded due to instability in aqueous environments. Only ReO$_4^-$, [B$_{12}$H$_{12}$]$^{2-}$, and [Co(7,8-C$_2$B$_9$H$_{11}$)$_2$]$^-$ proved compatible with immunostaining conditions without causing tissue destruction or precipitation. [B$_{12}$H$_{12}$]$^{2-}$/γCD produced the best staining sensitivity, specificity and signal homogeneity across depth (Supplementary Fig. 5), while the effect of derivatizing γCD was negligible (Supplementary Fig. 5). Finally, we chose the more soluble 2-hydroxypropylated derivative (2HPγCD) for its higher water solubility in our applications. We term our method INSIHGT, for <u>In situ</u> <u>host-</u>guest chemistry for <u>t</u>hree-dimensional <u>h</u>istology.

## In situ host–guest chemistry for three-dimensional histology (INSIHGT)

INSIHGT was designed to be a minimally perturbative, deeply and homogeneously penetrating staining method for 3D histology. Designed for affordability and scalability, INSIHGT involves simply incubating the conventional formaldehyde-fixed tissues in [B$_{12}$H$_{12}$]$^{2-}$/PBS with antibodies, then in 2HPγCD/PBS (Fig. 2a) - both at room temperature with no specialized equipment. We compared INSIHGT with other 3D IHC techniques using a stringent benchmarking experiment as previously published (see "Methods", Supplementary Fig. 4) to compare their penetration depths and homogeneity[6,9]. Briefly, a mouse hemibrain was first stained in bulk for an antigen using various deep immunostaining methods ("bulk-staining"), followed by cutting the tissue coronally in the middle (thickest dimension) and re-stained for the same marker with a different fluorophore using a standardized control method ("cut-staining"), which serves as the

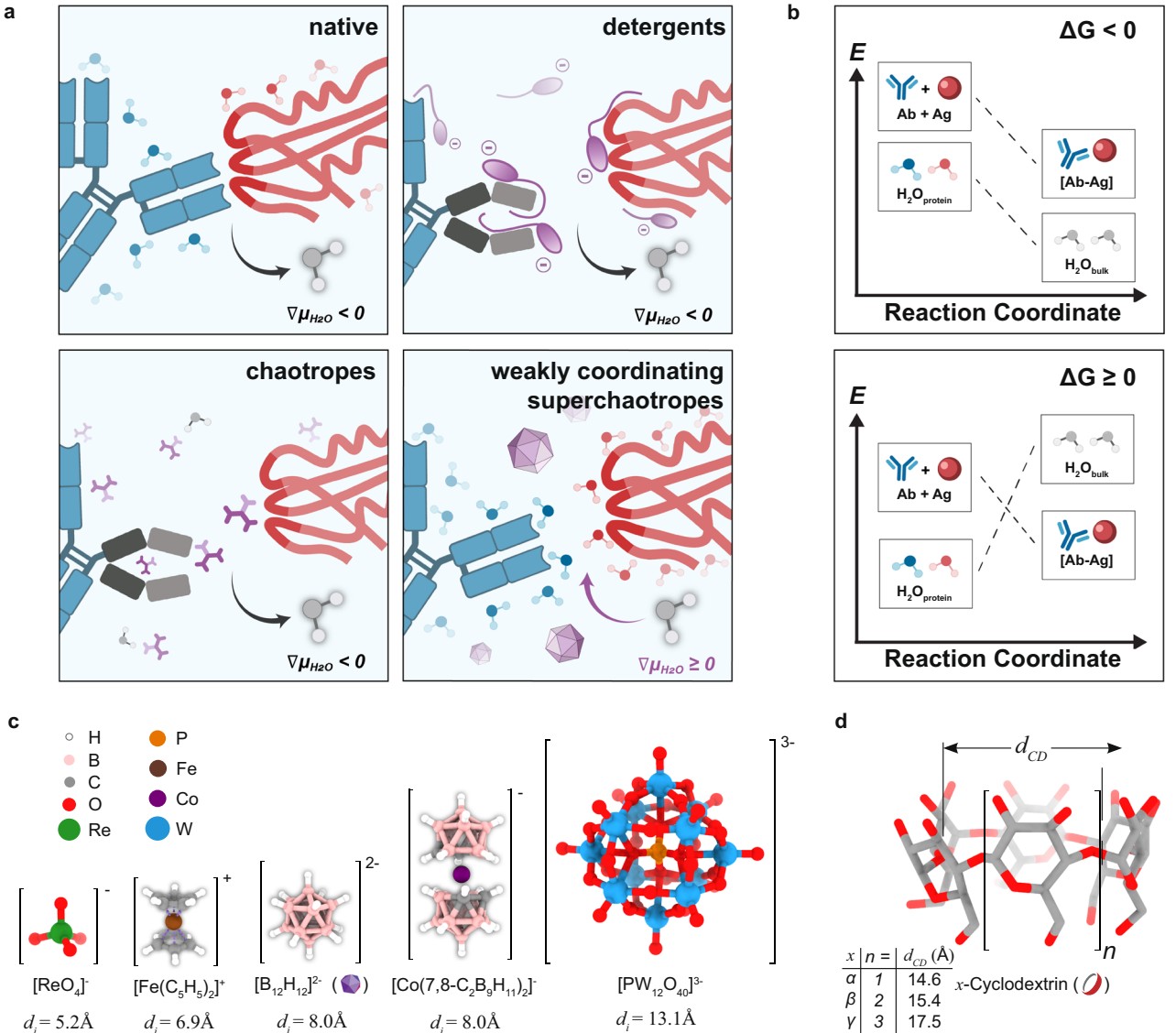

**Fig. 1 | INSIHGT conceptualization and key components.**
**a**, **b** Reconceptualization of the biomolecular binding phenomena. **a** Microscopic view. Native antibody-antigen binding (left upper, in an aqueous medium) requires the desolvation of their solvation shells ($\nabla\mu_{H2O} < 0$). Detergents (right upper) and chaotropes (left lower) solubilize proteins by masking binding sites (and displacing the solvent shell), but they may lead to protein denaturation in high concentrations (black arm of the antibody). Weakly coordinating superchaotropes (right lower,

WCS, e.g., [B$_{12}$H$_{12}$]$^{2-}$) instead solubilizes proteins by favoring solvent-protein interactions ($\nabla\mu_{H2O} \geq 0$), striking a balance between protein solubilization and stabilization. Created with Biorender.com. **b** Energetic view. Antibody-antigen binding without WCS (upper panel), occurs spontaneously with $\Delta G_{tot} < 0$, while with WCA (bottom panel), is unfavorable with $\Delta G_{tot} \geq 0$. Created with Biorender.com.
**c** Structures and ionic diameters ($d_i$) of weakly coordinating ions tested. **d** General structure of cyclodextrins and their cavity opening diameter ($d_{CD}$).

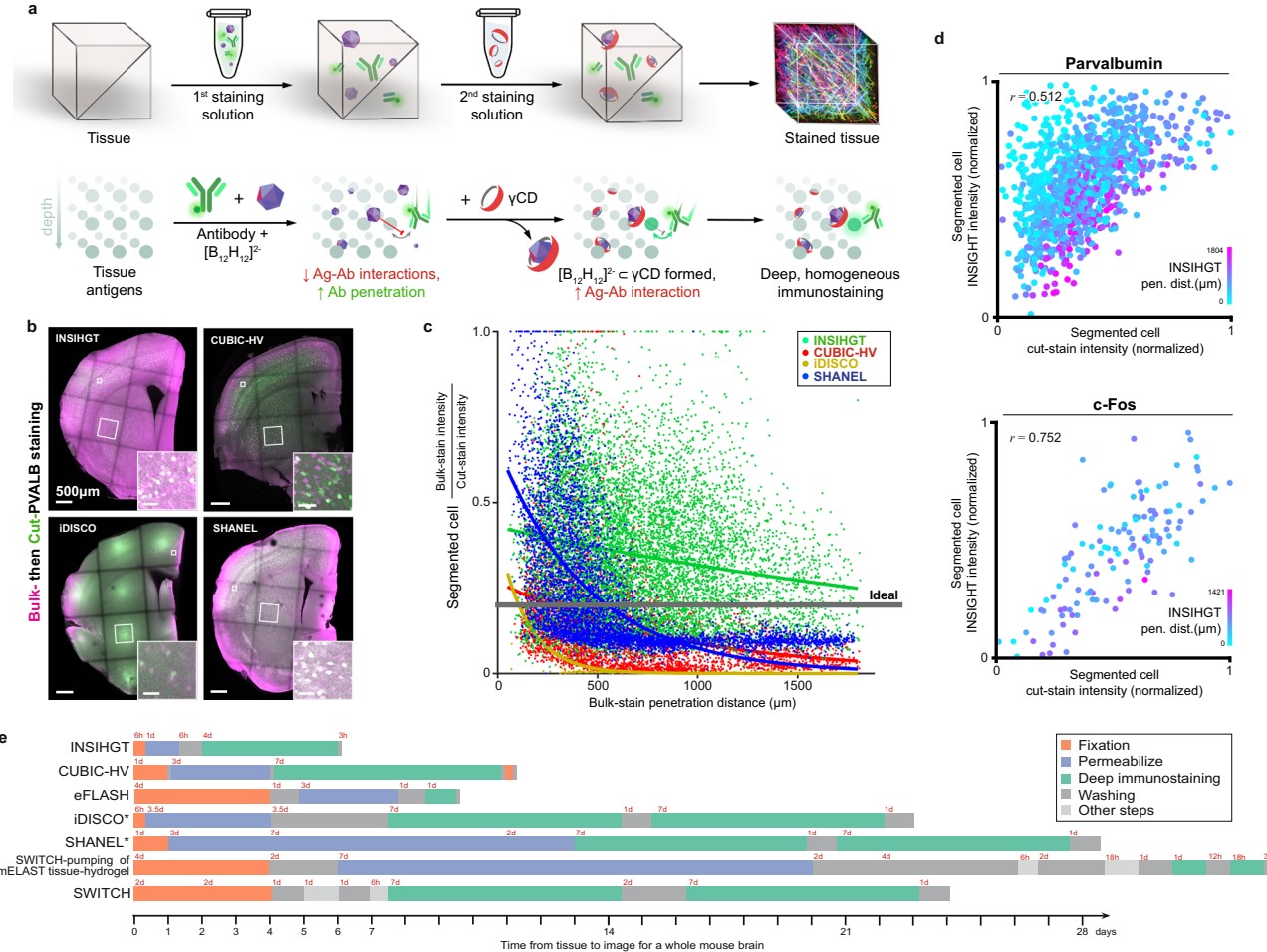

**Fig. 2 | Homogeneous and deep staining with INSIHGT. a** Experimental steps and principle of INSIHGT for immunostaining. Top row: Tissue is infiltrated with antibodies and a weakly coordinating superchaotrope ($[B_{12}H_{12}]^{2-}$, purple dodecahedron) in the 1st staining solution and then transferred into the 2nd solution containing a complexation agent (CD, red ring). Bottom row: The molecular principles of INSIHGT. Weakly coordinating superchaotropes prevents antibody-antigen interactions, removing penetration obstacles. After homogeneous infiltration, subsequent γCD infiltration complexes the $[B_{12}H_{12}]^{2-}$ ions, allowing deep tissue immunostaining. Reproduced with permission from Illumos Limited. **b** Benchmark results of four buffers used in deep immunostaining. Enlarged views of smaller areas are shown in insets. Parvalbumin (PVALB) immunostaining signals on cut surface: magenta, bulk-staining signal; green: cut-staining PVALB signal (refer to Supplementary Fig. 1). **c** Quantification of bulk:cut-staining signal ratio against penetration distance for segmented cells. Each dot represents a cell. Lines are single-term exponential decay regression curves. The signal decay distance constants (τ) are shown in Supplementary Table 1. Hypothetical ideal method performance is shown as a gray line (τ→0+). **d** Correlation of INSIHGT signal with reference (cut-staining intensity) signal, illustrating 3D quantitative immunostaining. *r*: Pearson correlation coefficient. **e** Timeline illustration for a whole mouse brain processing experiment with the different benchmarked methods (drawn to scale). *indicates methods where in principle the use of secondary antibody Fab fragments can lead to halved immunostaining times.

reference signal without penetration limitations. The tissue was then imaged on the cut face to compare the bulk-staining intensity (deep staining method signal) and cut-staining intensity (reference signal) as a function of the bulk-staining penetration depth. We found that INSIHGT achieved the deepest immunolabeling penetration with the best signal homogeneity throughout the penetration depth (Fig. 2b). To quantitatively compare the signal, we segmented the labeled cells and compared the ratio between the deep immunolabelling signal and the reference signal against their penetration depths. Exponential decay curve fitting showed that the signal homogeneity was near-ideal (Fig. 2c, Supplementary Table 1)—where there was negligible decay in deep immunolabelling signals across the penetration depth. We repeated our benchmarking experiment with different markers, and by correlating INSIHGT signal with the reference signal, we found INSIHGT provides reliable relative quantification of cellular marker expression levels throughout an entire mouse hemi-brain stained for 1 day (Fig. 2d). We supplemented our comparison with the binding kinetics modulating buffers employed in eFLASH and SWITCH-

pumping of mELAST tissue-hydrogel, as we lacked the specialized equipment to provide the external force fields and mechanical compressions, respectively (Supplementary Fig. 6). For SWITCH-pumping of mELAST tissue-hydrogel, we utilized the latest protocol and buffer recipe[13]. Our results also showed the use of binding kinetics modulating buffers alone from eFLASH and SWITCH-pumping of mELAST tissue-hydrogel lead to shallower staining penetration than INSIHGT, confirming the deep penetration of these methods is mainly contributed by the added external force fields and mechanical compressions, respectively. Hence, with excellent penetration homogeneity with a simple operating protocol, INSIHGT can be the ideal method for mapping whole organs with cellular resolution. It is also the fastest deep immunolabelling from tissue harvesting to image (Fig. 2e). Due to its compatibility with solvent-based delipidation methods, we recommend the use of solvent-based clearing[17–19] for an overall fastest INSIHGT protocol, although aqueous-based clearing techniques are also compatible (see "INSIHGT protocol in Supplementary Materials" for further discussions). However, protocols involving the use of

Triton X-100[8,15] and triethylamine[19] must be replaced with alternatives as they form precipitates with $[B_{12}H_{12}]^{2-}$.

Notably, after washing, only a negligible effect of $[B_{12}H_{12}]^{2-}$-treatment will remain within the tissue. This is evident as the cut-staining intensity profile of INSIHGT showed very steep exponential decay with increasing cut-staining penetration depth, and became similar to that of iDISCO (Supplementary Fig. 7) which has identical tissue pre-processing steps. Upon the addition of 2HPγCD and washing off the so-formed complexes, the penetration enhancement effect was completely abolished. This suggests that $[B_{12}H_{12}]^{2-}$ and cyclodextrins do not further permeabilize or disrupt the delipidated tissue.

## High-throughput, multiplexed, dense whole organ mapping

After confirming INSIHGT can achieve uniform, deeply penetrating immunostaining, we next applied INSIHGT to address the challenges in whole organ multiplexed immunostaining, where the limited penetration of macromolecular probes hinders the scale, speed, or choice of antigens that can be reliably mapped. Due to the operational simplicity, scaling up the sample size in organ mapping experiments with INSIHGT is straightforward and can be done using multiwell cell culture plates (Fig. 3a). For example, we demonstrated our case by mapping 14 mouse kidneys in parallel (Fig. 3b) within 6 days of tissue harvesting using a standard 24-well cell culture plate.

We then exemplify the capability of INSIHGT to simultaneously map densely expressed targets in whole organs (Fig. 3c-i, Supplementary Fig. 8-9). We first performed multiplexed staining on mouse kidney with 3 days of incubation for *Lycopersicon esculentum* lectin (LEL), Peanut agglutinin (PNA), *Griffonia simplicifolia* lectin (GSL), and AQP-1, which are targets associated with poor probe penetration due to their binding targets' dense expression (Fig. 3c-d, Supplementary Fig. 2, Supplementary Fig. 9a, b). With the use of INSIHGT, the dense tubules and vascular structures can be reliably visualized and traced (Supplementary Fig. 8).

We then proceeded to map the whole brain of a 3-year-old mouse at the time of euthanasia. We utilized INSIHGT with 3 days of staining for Calbindin (CALB1), NeuN, and c-Fos, providing cell type and activity information across the aged organ (Fig. 3e-i, Supplementary Fig. 9c). With whole organ sampling, we identified regions where aging-related changes were prominent, these include cavitations in the bilateral thalamus and striatum (Fig. 3g, h), as well as calbindin-positive deposits in the stratum radiatum of hippocampus (Fig. 3i). Interestingly, there seems to be an increased c-Fos expression level among the neurons surrounding thalamic cavitations (Fig. 3g) which are located deep within the brain tissue and thus cannot be explained by preferential antibody penetration, suggesting these cavitations may affect baseline neuronal activities. Similar 1-step multiplexed mapping of calcium-binding proteins across a whole adult mouse brain can also be performed with 3 days of staining (with a fixed tissue-to-image time of 6 days) (Fig. 3j–l, Supplementary Movie 1). Similarly, whole adult mouse brain mapping and statistics can be obtained for ~35 million NeuN+ cells, their GABA quantities and c-Fos expression levels using the same protocol (Supplementary Fig. 10), allowing structure, neurotransmitter, and activity markers to be analyzed simultaneously.

Overall, INSIHGT overcomes technical, operational, and cost bottlenecks towards accessible organ mapping for every basic molecular biology laboratory, providing rapid workflows to qualitatively evaluate organ-wide structural, molecular, and functional changes in health and disease, regardless of the spatial density of the visualization target.

## Boron cluster-based supramolecular histochemistry as a foundation for spatial multi-omics

With the maturation of single-cell omics technologies, integrating these high-dimensional datasets becomes problematic. Embedding these data in their native 3D spatial contexts is the most biologically informative approach. Hence, we next tested whether our boron cluster supramolecular chemistry allows the retention and detection of multiple classes of biomolecules and their features, based on which 3D spatial multi-omics technologies can be developed.

With identical tissue processing steps and INSIHGT conditions, we tested 357 antibodies and found 323 of them (90.5%) produced the expected immunostaining patterns as manually validated with reference to the human protein atlas and/ or existing literature (Fig. 4a, Supplementary Figs. 11–15, Supplementary Table 2). This was at least six times the number of compatible antibodies demonstrated by any other deep immunostaining method (Fig. 4a), demonstrating the robustness and scalability of INSIHGT. Antigens ranging from small molecules (e.g., neurotransmitters), epigenetic modifications, peptides to proteins and their phosphorylated forms were detectable using INSIHGT (Fig. 4b, c). The specificity of immunostaining even allowed the degree of lysine methylations (i.e., mono-, di- and tri-methylation) and the symmetricity of arginine dimethylations to be distinguished from one another (Fig. 4b). We further tested 21 lectins to detect complex glycosylations, proving that $[B_{12}H_{12}]^{2-}$ do not sequester divalent metal ions essential for their carbohydrate recognition (Fig. 4d, Supplementary Fig. 16).

Small molecule dyes such as nucleic acid probes, which are mostly positively charged, present a separate challenge as they precipitate with *closo*-dodecaborates, forming $[probe]^{n+}/[B_{12}H_{12}]^{2-}$ precipitates when co-applied with INSIHGT. We found size-matched and charge-complementing cyclodextrin derivatives as cost-effective supramolecular host agents for non-destructive deep tissue penetration and preventing precipitation. For example, sulfobutylether-βCD (SBEβCD) (Fig. 4e) can react with nucleic acid probes to form [probe⊂SBEβCD], which exhibits penetration enhancement during INSIHGT (Fig. 4f, g) without precipitation problems. The so-formed [probe⊂SBEβCD] complex can thus be co-incubated with antibodies in the presence of $[B_{12}H_{12}]^{2-}$ for a simpler protocol.

We also performed RNA integrity number (RIN) and whole genome DNA extraction analyses on INSIHGT-treated samples (Supplementary Fig. 17). We found each step of the INSIHGT protocol did not result in a significant decrease in RNA integrity number (RIN) (Supplementary Fig. 17a). The total RNA extracted after undergoing the whole INSIHGT protocol has an RIN of 7.2, compared with a RIN of 9 from a treatment-naive control sample. For whole genome DNA, both control versus INSIHGT-protocol-treated samples have similar sample integrity and total DNA yield per mm³ sample (14.6 μg versus 10.12 μg), as well as subsequent whole genome sequencing quality (total clean base 114.5 Gb versus 125.2 Gb) with both having a mapping rate of 99.96% (Supplementary Fig. 17b see also "Methods" on the quality control descriptions). With RNA sequencing whole transcriptomic comparing an INSIHGT-treated sample and a paired control sample (the opposite mouse hemibrain), the results showed essentially no differentially expressed genes profiles (Supplementary Fig. 17c). The Pearson correlation coefficient of the expression of all genes was 0.967. Hence, unsurprisingly, we found single-molecule fluorescent in situ hybridization (FISH) is also applicable for co-detection of protein antigens and RNAs with INSIHGT. Combining all the above probes, simultaneous 3D visualization of protein antigens, RNA transcripts, protein glycosylations, epigenetic modifications, and nuclear DNA is possible using a mixed supramolecular system in conventionally formalin-fixed intact tissue (Fig. 4h, Table 1). Taken together, our results suggest in situ boron cluster supramolecular histochemistry can form the foundation for volumetric spatial multi-omics method development. The implication of well-preserved RNAs suggests the possibility of post-INSIHGT section-based spatial transcriptomics.

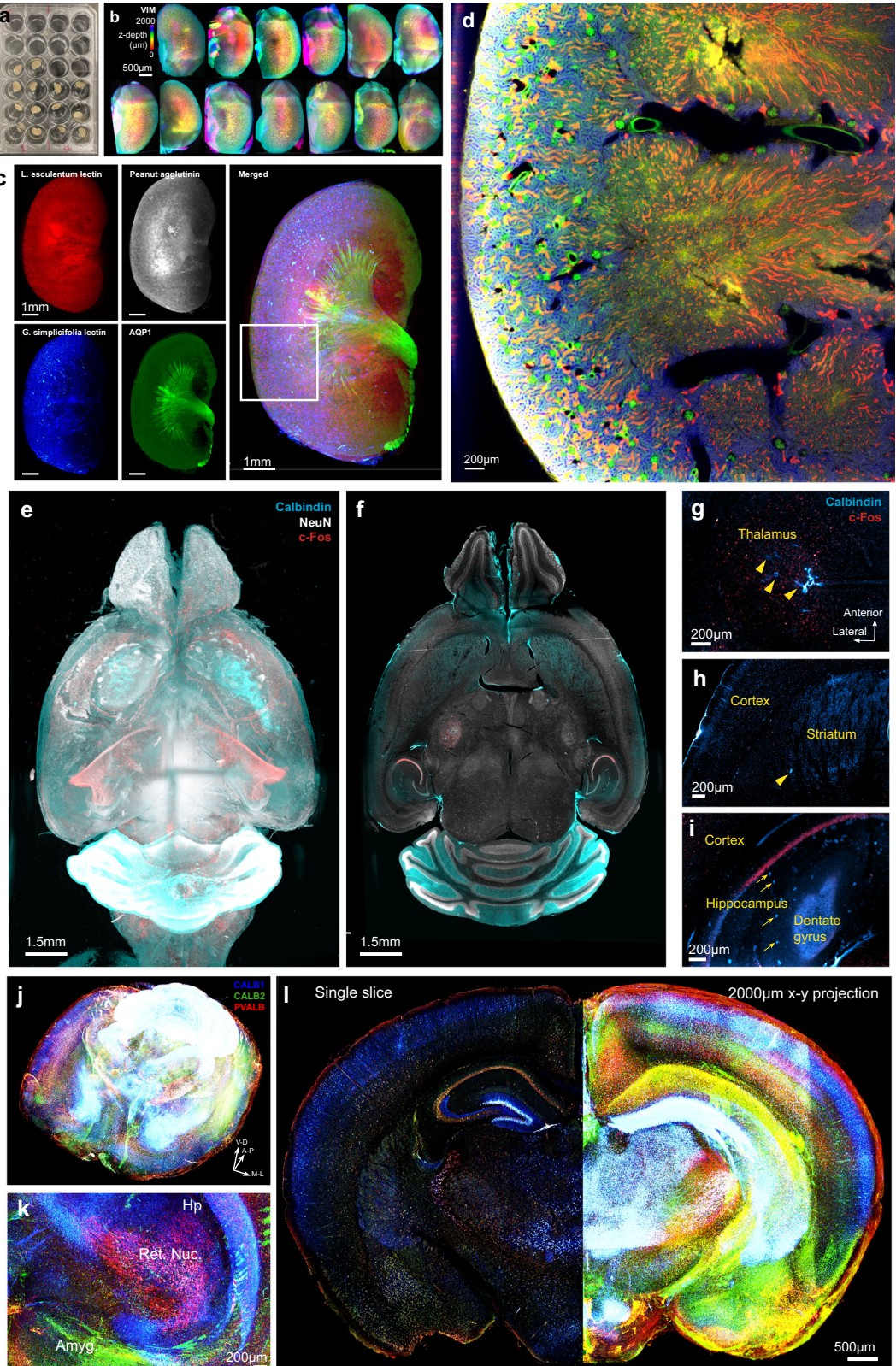

## Centimeter-scale 3D histochemistry by isolated diffusional propagation

Since antibody penetration remains the most challenging obstacle, we focus the remainder of our investigation on larger-scale 3D immunophenotyping. We thus applied INSIHGT to visualize centimeter-scale human brain samples, without using any external force fields to drive the penetration of macromolecular probes.

These large, pigmented samples were sliced in the middle of the tissues' smallest dimensions to allow imaging of the deepest areas with tiling confocal microscopy. We show that INSIHGT can process a 1.5 cm × 1.5 cm × 3 cm human cortex block for parvalbumin (PV) (Fig. 5a–c), with excellent homogeneity and demonstration of parvalbumin neurons predominantly in layer 4 of the human cortex.

**Fig. 3 | High-throughput whole-organ deep immunostaining with dense mapping using INSIHGT. a** Parallelized sample processing with INSIHGT, exemplified with whole mouse kidneys. **b** Whole organ imaging results with parallelized INSIHGT for the samples shown in (**a**), showing vimentin INSIHGT signals color-coded by z-depth. One sample was dropped due to manual errors. **c,d** Whole mouse kidney densely multiplexed visualization with *Lycopersicon esculentum* lectin (red), Peanut agglutinin (gray), *Griffonia simplicifolia* lectin I (blue), and AQP-1 (green). **d** Enlarged 2D view of the white boxed area in (**c**). **e, f** Rendered view of whole mouse brain multiplexed Calbindin, NeuN, and c-Fos mapping of a 3-year-old mouse. **g–i** Age-related structural and molecular changes in the thalamus (**g**) and striatum (**h**) with cavitations (indicated by yellow arrowheads), and the hippocampus (**i**) with CALB1-positive deposits (indicated by yellow arrows). **j** Whole brain multiplexed staining of the calcium-binding proteins calbindin (CALB1), calretinin (CALB2), and parvalbumin (PVALB) with 3 days of INSIHGT staining. **k** Zoomed in 3D rendering view on the hippocampus (Hp), reticular nucleus of thalamus (Ret. Nuc.), and amygdala (Amyg.). **l** A single coronal slice view (left) and a 2 mm-thick anteroposterior projection (right) of the same sample.

We then scaled INSIHGT to a 1.75 cm × 2.0 cm × 2.2 cm human cerebellum block for blood vessels (using *Griffonia simplicifolia lectin I, GSL-I*) (Fig. 5d–f). As light-sheet microscopy is suboptimal due to the large human sample, we assessed the INSIHGT staining penetration on the cut face along the thickest dimension using confocal microscopy (Fig. 5e, Supplementary Fig. 18). This again reveals excellent homogeneity with no decay of signal across the centimeter of penetration depth. This shows that the use of boron cluster-based host-guest chemistry remains applicable for highly complex environments at the centimeter scale. The results further show that macromolecular transport within a dense biological matrix can remain unrestricted in a non-denaturing manner by globally adjusting inter-biomolecular interactions.

We further applied INSIHGT to a 1.0 cm × 1.4 cm × 1.4 cm human brainstem with dementia with Lewy bodies (DLB) for phosphorylated alpha-synuclein at serine 129 (αSyn-pS129) (Fig. 5g-i, Supplementary Fig. 19). The large scale of imaging enabled registration and hence correlation with mesoscale imaging modalities such as magnetic resonance imaging (MRI) (Fig. 5g, Supplementary Movie 2). With this, we confirmed the localization of Lewy body pathologies to the locus ceruleus complex–subcerulean nuclei[27] and substantia nigra, in keeping with the prominent rapid eye movement sleep behavior disorder (RBD) symptoms of this patient. Such a radio-histopathology approach would allow for correlative structural-molecular studies for neurodegenerative diseases. Overall, the capability of INSIHGT in achieving centimeter-sized tissue staining bridges the microscopic and mesoscopic imaging modalities, providing a general approach to correlative magnetic resonance-molecular imaging.

**Volumetric spatial morpho-proteomic cartography for cell type identification and neuropeptide proximity analysis**

We next extended along the molecular dimension on conventionally fixed tissues, where highly multiplexed immunostaining-based molecular profiling in 3D had not been accomplished previously. A single round of INSIHGT-based indirect immunofluorescence plus lectin histochemistry can simultaneously map up to 6 antigens (Supplementary Fig. 20), tolerating a total protein concentration at >0.5 μg/μl in the staining buffer, and is limited only by spectral overlap and species compatibility. To achieve higher multiplexing, antibodies can be stripped off with 0.1 M sodium sulfite in the $[B_{12}H_{12}]^{2-}$-containing buffer after overnight incubation at 37 °C (Fig. 6a, Supplementary Fig. 21). Since $[B_{12}H_{12}]^{2-}$ does not significantly disrupt intramolecular and intermolecular noncovalent protein interactions, the approach can be directly applied to routine formaldehyde-fixed tissues, we observed no tissue damage and little distortion, obviating the need for additional or specialist fixation methods.

We exemplified this approach by mapping 28 marker expression levels in a 2 mm-thick mouse hypothalamus slice over 7 imaging rounds (Fig. 6a–c, Supplementary Figs. 22, 23). With each iterative round taking 48 h (including imaging, retrieval and elution), the whole manual process from tissue preparation to the 28-plex image took 16 days. After registration and segmentation using Cellpose 2.0[28] (Fig. 6d, e, see "Methods"), we obtained 192,075 cells and their differentially expressed proteins (DEPs) based on immunostaining signals. Note that other user-friendly approaches such as StarDist[29] and

BCFind[30] can also be used. Omitting 3 blood vessel channels, we then obtained the normalized mean intensities of the remaining 25 markers, their standard deviations (S.D.s) of signal intensities of the same 25 markers, as well as their distance to the nearest vessels for dimensionality reduction analysis and clustering. The S.D.s of signal intensities for each cell served as a measure of heterogeneous expression of a certain marker within the cell (e.g., strictly cytoplasmic or nuclear expression will have a higher S.D. than a marker expressing in both the cytoplasms and nuclei, as illustrated in Fig. 6e). Uniform manifold approximation and projection (UMAP) analysis of a subset of 84,139 cells based on these 51 markers (Fig. 6f, Supplementary Figs. 24, 25) plus their distance to the nearest vessels revealed 42 cell type clusters, allowing their 3D spatial interrelationships to be determined (Supplementary Fig. 26).

INSIHGT allows both 3D morphology and molecular information to be well-visualized via immunostaining, which is more difficult to access via current section-based spatial transcriptomics or single-cell multi-omics despite ongoing efforts[31]. Recent characterizations of neuronal network activities based on the diffusional spread of neuropeptides highlight the need for 3D spatial mapping of protein antigens. To obtain these morphological-molecular relationships using INSIHGT, we segmented the neuropeptide Y (NPY)-positive fibers and computed the 3D distance to each UMAP-clustered cell types' somatic membrane (Fig. 6g–j). While most clusters have a similar distance from NPY fibers, certain clustered cells (notably right tile clusters 1 and 2) are more proximally associated with NPY fibers, suggesting these cell clusters are differentially modulated by NPY when isotropic diffusion is assumed in the local brain parenchyma. Nonetheless, our dataset and analysis demonstrated it is possible to estimate the likely modulatory influence for a given cell-neuropeptide pair, providing an alternative approach to discovering neuronal dynamics paradigms.

**Fine-scale 3D imaging reveals unsuspected intercellular contacts traversing the Bowman space in mouse kidneys**

We found that the process of INSIHGT from fixation to completion preserves delicate structures such as free-hanging filaments and podia, enabling fine-scale analysis of compact structures such as the renal glomeruli. We found unsuspected intercellular contacts traversing the Bowman space, which was not known to be present in normal glomeruli even with serial sectioning electron microscopy studies[32–36] (Fig. 7a, b). These filaments are mostly originated from the podocytic surface, although some were also seen to emerge from PECs. They were numerous and found around the glomerular globe (Fig. 7c), with varied in their length, distance from each other, and morphologies (Fig. 7d, Supplementary Fig. 27).

We classified these podocyte-to-PEC microfilaments into "reachers" and "stayers", depending on whether they reached the PEC surface or not (Fig. 7e). Microfilaments of the reachers-type were more numerous than the stayers-type per glomerulus (Fig. 7f). Visually, we noted the emergence of these filaments tended to cluster together, especially for the reachers-type. To quantify such spatial clustering, we calculated the glomerular surface geodesic distances between the podocytic attachment points for each microfilament, which showed an inverse relationship with their path lengths (Fig. 7g), and reachers-type filament are geodesically located

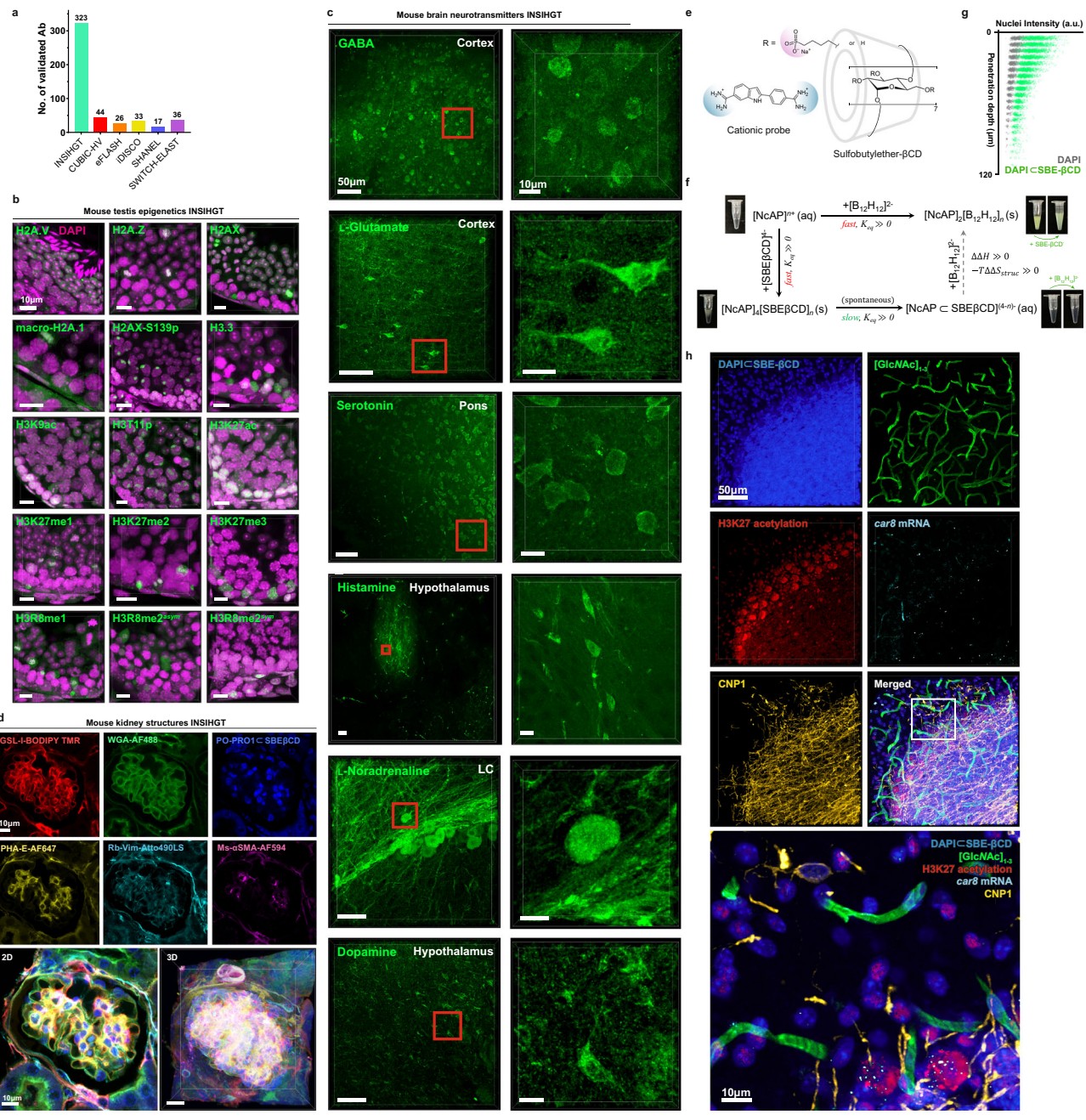

**Fig. 4 | Multi-modality INSIHGT for generality. a** Number of validated antibodies compatible with each benchmarked method. **b–d** INSIHGT's compatibility in revealing the 3D location of various features. **b** Epigenetic markers based on histone post-translational modifications and isoforms. **c** Neurotransmitters. LC: locus ceruleus. **d** Structural features, using one-step multiplexed supramolecular histochemistry, multiplexed supramolecular lectin histochemistry, and the supramolecular dye complex [PO-PRO1⊂SBEβCD]. **e** Nucleic acid probe (NAP, DAPI structure shown) complexation by SBEβCD for improved tissue penetration.

**f** Thermodynamic scheme of NAP's complexation reaction with SBEβCD. SBEβCD neither redissolves DAPI/[B₁₂H₁₂]²⁻ precipitates nor precipitates DAPI out from the [DAPI⊂SBEβCD] complex, suggesting kinetic stabilization. **g** Quantification of penetration depths of [DAPI⊂SBEβCD] compared to traditional DAPI staining. **h** Multimodal 3D molecular phenotyping in a 1 mm-thick mouse cerebellum slice for proteins (CNP1), nucleic acids (*car8*, DAPI), epigenetic modifications (H3K27 acetylation), and glycosylations ([GlcNAc]₁₋₃) with INSIHGT. Imaging was limited to 200 µm due to working distance constraints.

## Table 1 | Sequences of FISH HCR probes applied

| *car8* (with B3 Hairpin initiator) |
| --- |
| GTCCCTGCCTCTATATCTCCACTCAACTTTAACCCG TACAA GCTTCTCTGGAGTTTAGGTTGATAGG TAAAA AAAGTCTAATCCGTCCCTGCCTCTATATCTCCACTC |
| GTCCCTGCCTCTATATCTCCACTCAACTTTAACCCG TACAA CTTCATACAGCTCAAACTCCTGTCC TAAAA AAAGTCTAATCCGTCCCTGCCTCTATATCTCCACTC |
| GTCCCTGCCTCTATATCTCCACTCAACTTTAACCCG TACAA GCTTTGAAATTGACCGTGTGCTCAGA TAAAA AAAGTCTAATCCGTCCCTGCCTCTATATCTCCACTC |

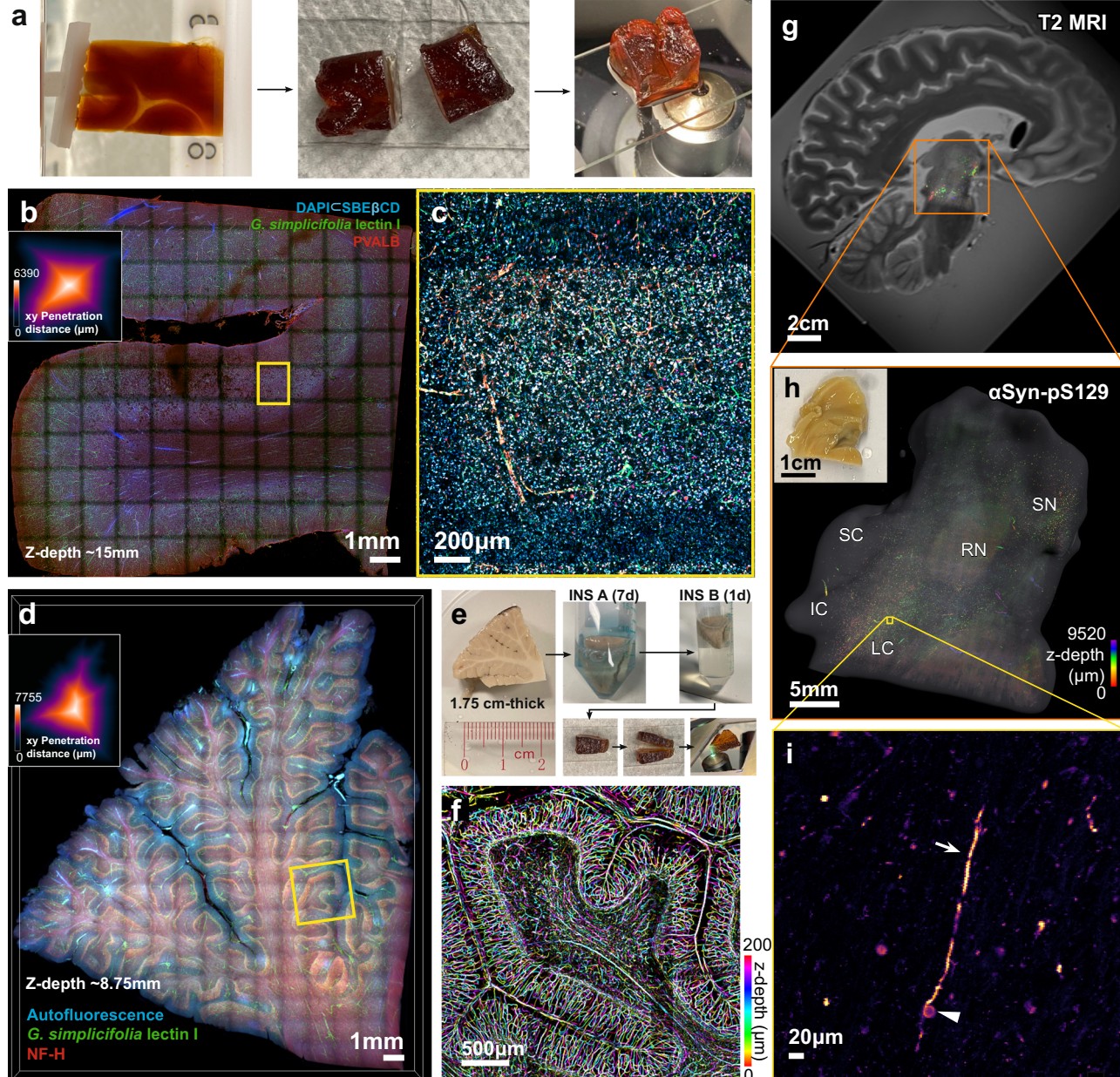

**Fig. 5 | Centimeter-scale INSIHGT. a** A human cortex block (1.5 cm × 1.5 cm × 3 cm), processed with INSIHGT and cut in the middle for confocal imaging to confirm penetration depth. **b** Confocal tiled image of the cut face from (**a**), stained with [DAPI⊂SBEβCD], *G. simplicifolia* lectin I (GSL-I), and for parvalbumin (PVALB). Inset: penetration depth over the imaging surface. **c** Enlarged view of the yellow boxed area in (**b**). **d** A human cerebellum tissue block (1.75 cm × 2.0 cm × 2.3 cm) processed with INSIHGT, stained with GSL-I lectin and NF-H. Inset: penetration depth over the imaging surface. **e** Illustrated tissue processing steps: tissue is stained for 7 days in INSIHGT buffer A (with $[B_{12}H_{12}]^{2-}$), washed in INSIHGT buffer B (with

2HPγCD), and sliced perpendicular to the thinnest dimension at the midpoint. **f** Enlarged view of the yellow boxed area in (**d**). **g** T2-weighted magnetic resonance imaging (MRI) of a patient's brain with dementia with Lewy bodies (DLB). A hemi-brainstem region spanning the pons to substantia nigra was stained for phosphorylated α-synuclein at serine 129 (αSyn-pS129). **h** αSyn-pS129 staining intensity color-coded in z-depth. Inset shows the specimen photograph. LC: locus ceruleus complex, IC inferior colliculus, RN red nucleus, SC superior colliculus, SN substantia nigra. **i** Zoomed-in view of the yellow boxed area showing Lewy neurites (arrow) and Lewy bodies (triangle).

nearer to each other than the stayers type (Supplementary Fig. 28). This suggests that the emergence of long, projecting microfilaments that reach across the Bowman space is localized on a few hotspots of the glomerular surface. Whether these hotspots of long-reaching microfilaments are driven by signals originated from the podocyte, the glomerular environment underneath, or the nearest PECs across the Bowmann space remains to be investigated and may reveal previously unsuspected podocyte physiological responses within their microenvironments. Notably, similar structures have been observed in the pathological state of cresenteric

glomerulonephritis, in conjunction with whole cells traversing the Bowman space. As cresenteric glmoerulonephiritis is a final common pathway of glomerulonephropathies, it would be interesting to investigate whether there is a continuum of progressive changes from microfilaments physiologically to larger trans-Bowman space connections pathologically. In addition, morphologically similar structures have been observed in the microglia[37], pericytes[38], between tumor and immune cells[39], and between normal and apoptotic cells in cell culture[40]. The podocyte-PEC connections described here thus add another organ to the growing list of

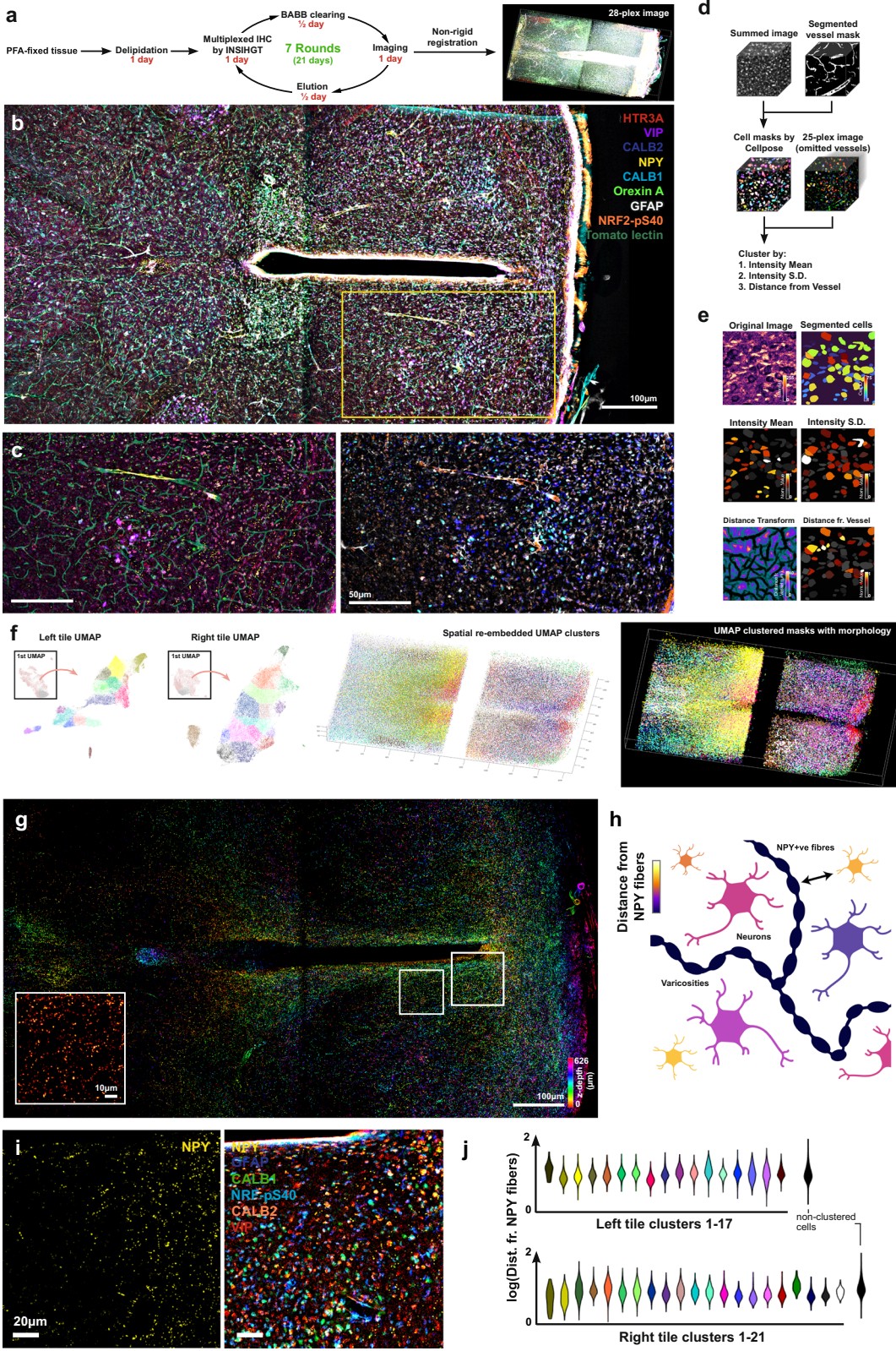

nanostructural connections mediating information and matter exchange between different cell types in their physiological states.

### Sparsely distributed neurofilament inclusions unique to the human cerebellum

We next completely mapped a 3 mm-thick (post-dehydration dimensions) human cerebellar folium for NF-H, GFAP, and blood vessels

(Fig 8a, Supplementary Figs. 29, 30, Supplementary Movie 3), with preserved details down to the Bergmann glia fibers, perivascular astrocytic endfeet, and Purkinje cell axons that make the amenable to 3D orientation analysis and visualization (Fig. 8b-d, Supplementary Figs. 29, 30). The detailed visualization of filamentous structures throughout the 3 mm-thickness is in stark contrast to our previous attempts with similar specimens employing various methods, which

**Fig. 6 | Multi-round multiplexed INSIHGT. a** Schematics of the processing steps for a 1mm-thick mouse hypothalamus sample. **b** A selection of multi-round immunostaining signals (for nine targets) displayed for the multi-round multiplexed INSIHGT-processed sample. **c** Enlarged view of the yellow boxed area in (**b**) with a complementary panel of markers. **d** Schematics of the image analysis process. **e** Illustrated results of a segmented image subset. The images below show corresponding cell segmentation and quantification results. **f** Left panel: Nested UMAP embedding of all segmented cells from both tiles of the image stack. Middle panel: The spatial locations of the different color-coded clusters. Right panel: Similar to the middle panel but with cellular morphology. **g** Color-coded z-projection of neuropeptide Y (NPY) staining signals. A higher magnification view for the left white boxed area is shown in the inset. **h** Schematic representation of distance measurement from NPY fibers to cell bodies via distance transformation. Created with Biorender.com. **i** NPY signal in a periventricular region (right white boxed area in **g**) is shown in the left panel, and with selected markers staining in the right panel. **j** Quantification of distance from NPY-expressing fibers for each cell type shown in violin plots, based on the 3D spatial locations of somas and NPY fibers.

showed weak NF-H signal in cerebellar sulci and barely visible GFAP signal in cerebellar white matter due to poor antibody penetration.

We discovered sparsely distributed NF-H-intense inclusions that are easily missed in 2D sectioning and thus remain poorly characterized. We manually traced and identified 1078 inclusions throughout the entire imaged volume (Fig. 8e, f), where they were found in all of the three basic layers of the cerebellar cortex. A typical morphology of one type of these inclusion is a single bright globular inclusion at the sub-Purkinje layer radial location, with an elongated thick fiber extension that coils back and project to the adjacent molecular layer (Fig. 8e). However, much more protean morphologies also exist (Fig. 8e, f, Supplementary Fig. 30). To capture the morphological and spatial diversities of these inclusions, we obtained their spatial-morphometric statistics (Supplementary Fig. 31a), followed by principal component analysis of the compiled morphometrics such as Sholl analysis and Horton-Strahler number. The results reveal most of these inclusions to be morphologically homogeneous with variations explained largely by their path lengths, with a small subset characterized by much higher branching of the NF-H-intense filaments (Supplementary Fig. 31b). However, further understanding of these inclusions awaits broader investigations in normal and various disease states other than in DLB. Preliminarily, we have also observed these inclusions in normal human cerebellum tissues (Supplementary Fig. 31c). With the advancements in technologies, correlated mulit-pronged approaches using superresolution microscopy, electron microscopy and spatially resolved proteomics are expected to help greatly clarify the pathobiology of these inclusions.

### INSIHGT bridges the gap between 3D histology and traditional 2D pathology in current clinical practice

The bio-orthogonal nature of the INSIHGT chemical system underlies its non-destructiveness. To highlight the clinical impact of INSIHGT in addition to 3D imaging of human samples, we found that INSIHGT-processed samples can be retrieved and processed as naïve tissues for traditional 2D histology via paraffin wax embedding and sectioning. Notably, staining qualities of routine hematoxylin and eosin (H&E) and various special stains on the post-INSIHGT processed slides were indistinguishable from the pre-INSIHGT processed slides even by a senior pathologist (Fig. 8g, h). In addition to not interfering with downstream clinical processes, the preserved quality of special staining allows for multi-modal cross-validation of 3D fluorescent imaging findings, making INSIHGT the ideal platform choice for next-generation histopathology (Fig. 8i). Together with the possibility for post-INSIHGT DNA and RNA sequencing, we envision (Supplementary Fig. 17) quantitative 3D information within clinical specimens can be maximally extracted and preserved with high authenticity in a non-consumptive manner using INSIHGT, and its fast speed promises compatibility with current clinical workflows and constraints, allowing digital pathology and precision medicine to benefit from 3D analysis.

### Discussion

The convergence of multiple technological advances has paved the way for the acquisition of large-scale molecular phenotyping datasets at single-cell resolution, most notably single-cell transcriptomics[36]. With a large number of previously undiscovered cell states, the quest to extend towards spatially resolved cell phenotyping based on translated and post-translationally expressed biomolecular signatures is paramount to understanding their structural and functional properties in biology[41].

Scalable, high-resolution 3D tissue mapping provides a powerful approach to further our understanding of these previously unidentified cell types. Clinically, 3D histology has been shown to improve diagnosis in bladder cancer[42], predict biochemical recurrence in prostate cancer[43], and evaluate response to chemotherapy in ovarian carcinoma[42]. By sampling across whole intact samples, 3D histology can deliver unbiased, quantitative, ground-truth data on the spatial distributions of molecules and cell types in their native tissue contexts[44]. However, 3D tissue imaging is yet to be widely adopted despite the increasing accessibility of tissue clearing, optical sectioning microscopy, and coding-free image processing software. This is in large part due to the limited penetration of probes that plague the field regardless of the combinations of these technologies employed[6,10], yielding variable, surface-biased data with questionable representativeness. Creative approaches have provided solutions to the penetration problem but are limited in their scalability and accessibility[6].

Constrained by the requirements of non-advective approaches and compatibility with off-the-shelf reagents, the development of INSIHGT involved re-examining biomolecular transport and protein stability from the first principles, which led us to identify weakly coordinating superchaotrope and its chemical activity modulation by in situ host–guest reactions to implement our theoretical formulation. With the use of *closo*-dodecaborate and cyclodextrin as an additive in PBS, we solved the bottleneck of 3D histology by providing a cost-efficient, scalable, and affordable approach to quantitatively map multiple molecules in centimeter-sized tissues. With an equivalent tissue processing pipeline to iDISCO[17], INSIHGT shares the same affordability and scalability while providing much faster processing and greatly improved image quality, due to enhanced antibody penetration depth and homogeneity. Mapping tissue blocks simultaneously in multi-well dishes is easily accomplished in any basic molecular biology laboratory. Such simplicity in operation makes it highly accessible and automatable, as it requires no specialized equipment or skills. Furthermore, cocktails of off-the-shelf antibodies can be directly added to PBS supplemented with $[B_{12}H_{12}]^{2-}$. Finally, we note that both $[B_{12}H_{12}]^{2-}$ salts and cyclodextrins are non-hazardous and stable indefinitely at ambient temperatures[16].

With the affordability and accessibility of INSIHGT, we anticipate its diverse applications in 2D and 3D histology applications. Meanwhile, boron cluster-based supramolecular histochemistry can form the backbone for 3D spatial molecular-structural-functional profiling methods and studies, as well as atlas mapping efforts. The high-depth, quantitative readout of well-preserved tissue biomolecules offered by INSIHGT forms the foundation for multiplexed, multi-modal, and multi-scale 3D spatial biology. By making non-destructive 3D tissue molecular probing accessible, INSIHGT can empower researchers to bridge molecular-structural inferences from subcellular to the organ-wide level, even up to clinical radiological imaging scales for radio-histopathological correlations. Finally, the compatibility of INSIHGT with downstream traditional 2D histology methods indicates its non-interference with subsequent clinical decision-making. This paves the

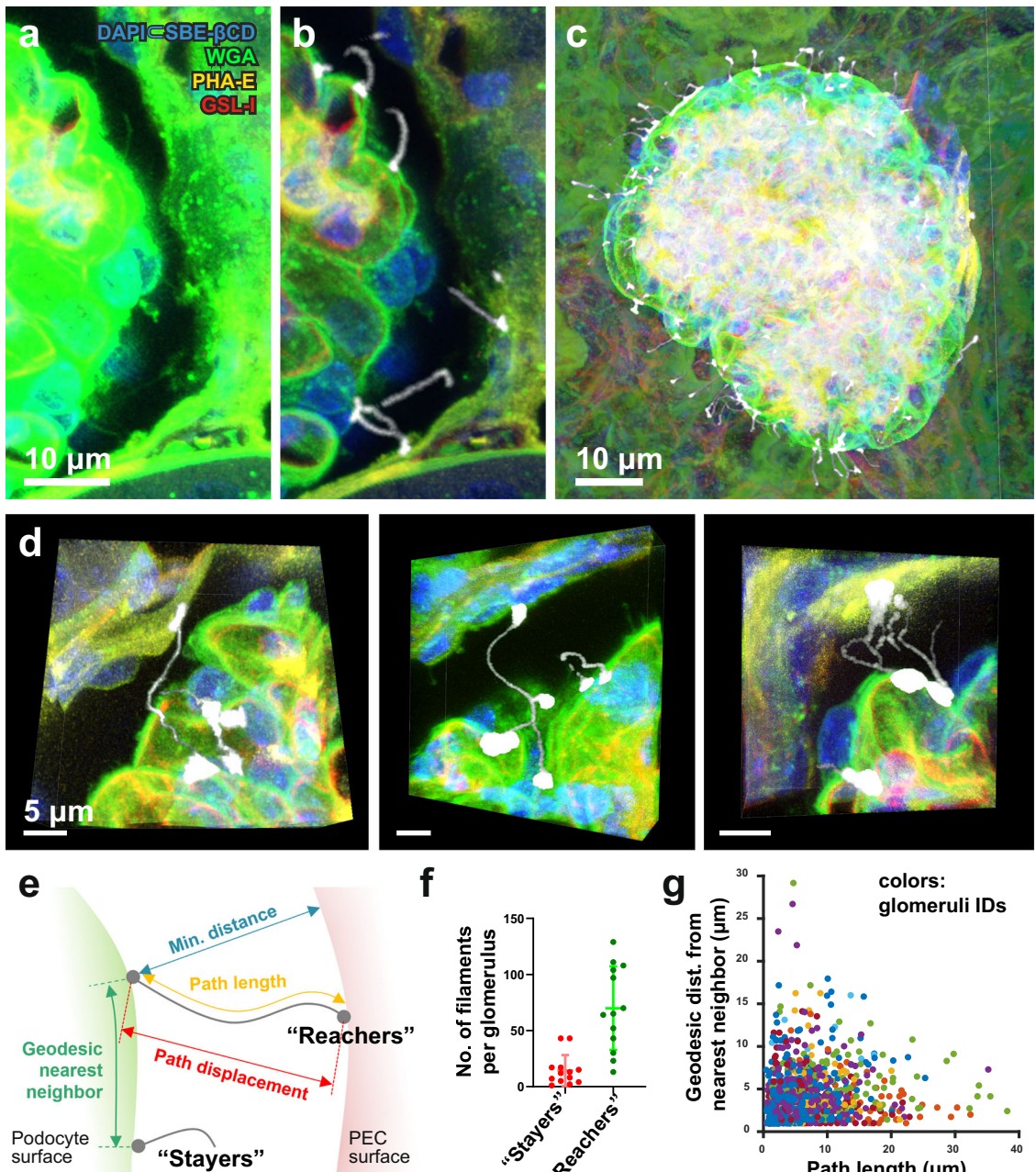

**Fig. 7 | INSIHGT reveals previously undescribed intercellular filaments traversing the Bowman space in mouse kidneys. a** Original image of multiplexed image of whole mouse glomerulus with full Bowman capsule. WGA Wheat germ agglutinin, PHA-E *Phaseolus vulgaris* hemagglutinin, GSL-I *Griffonia simplicifolia* lectin I. **b** Segmentation of microfilaments with en bloc preservation of native morphologies and spatial relationships in 3D Euclidean space. **c** Global representation of the 3D spatial distribution of microfilaments across the entire Bowman space. **d** Distinct and protean morphologies of the podocyte-to-parietal epithelial cell (PEC) microfilaments. **e** Schematic representation of reachers (contacting PEC surface) and stayers (remaining in the Bowman space) originating from podocyte surfaces with the related physical parameters. **f** Descriptive statistics of microfilament subtypes per analyzed glomeruli. $N = 14$ glomeruli analyzed across four mice (where three mice has four glomeruli imaged, one mouse has two glomeruli imaged, and the results were plotted together). **g** Correlation between the path length of each microfilament and the geodesic distance between its podocytic attachment point and its nearest neighbor. The data points were color-coded based on their glomerulus of origin in the dataset. Source data are provided as a Source Data file.

way for the translation and development of 3D histology-based tissue diagnostics, promising rapid and accurate generation of groundtruth data across entire tissue specimens.

We recognize that INSIHGT still has room for further improvements. Immunostaining penetration homogeneities for larger tissues and denser antigens can be further enhanced, Practically, this is limited to a maximum of ~2 cm³ sized tissues, and extremely dense antigens such as GAPDH, type I collagen, actin, and myosin remain difficult for whole organ staining with homogeneous penetration. Nonetheless, for

any antigens stained using the iDISCO+ protocol[45] with 7 days of primary antibody staining, INSIHGT with 3 days of antibody staining will at least provide 10–20× penetration enhancement, along with a noticeable enhancement in penetration homogeneity. Penetration can be further enhanced by prolonging the incubation times and ensuring an adequate amount probes has been added relative to the tissue expression level (see "Supplementary Note and INSIHGT protocol therein"). If available, the use of primary nanobodies with fluorescently-labeled secondary whole IgGs will further increase the penetration by about 5–10 times. In

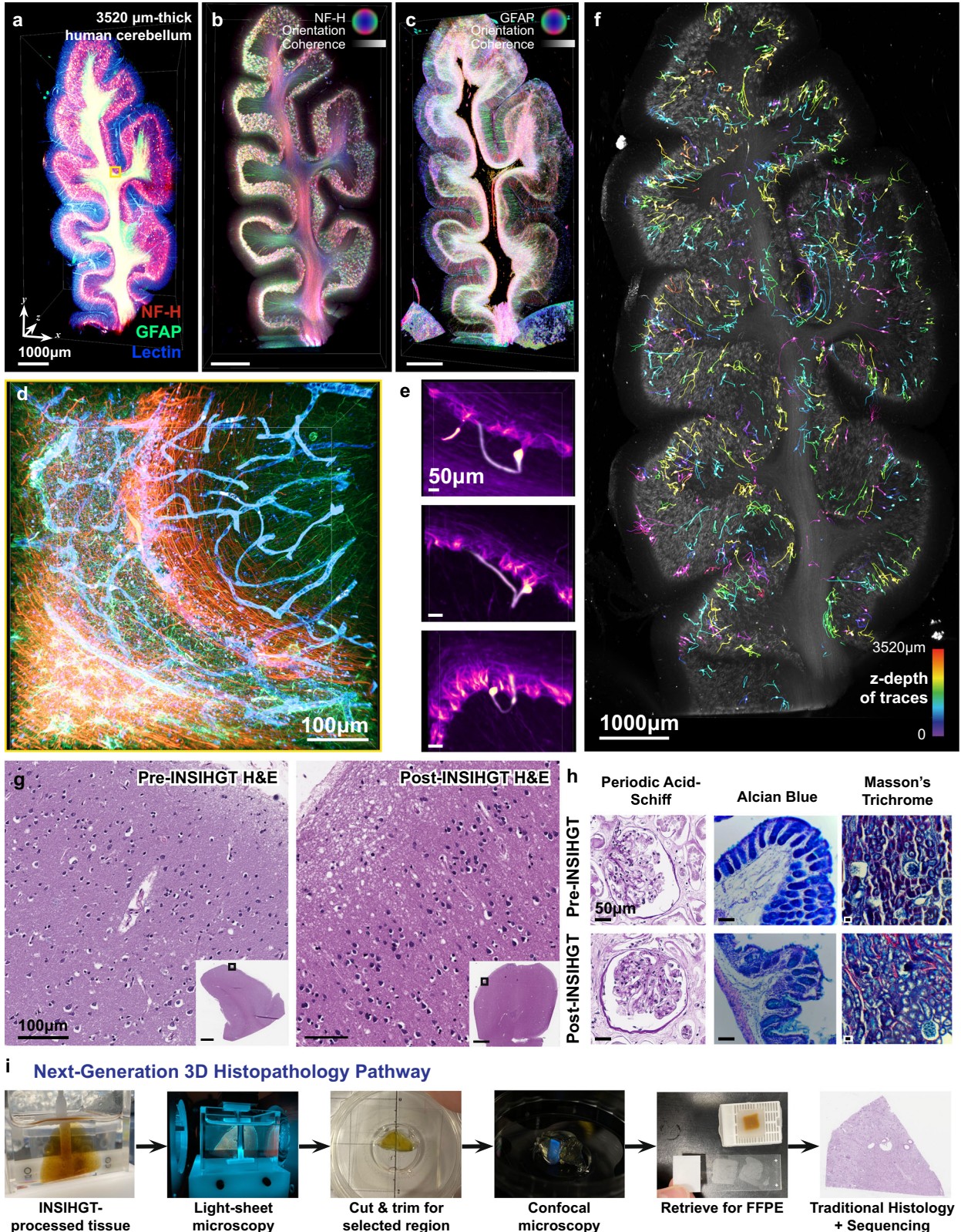

addition, the penetration homogeneity of small molecule dyes and lectins were still suboptimal for millimeter-scale tissues and remains to be further enhanced. In multi-round immunostaining, we noticed that the staining specificity and sensitivity deteriorated with each round of antibody elution with sulfite or β-mercaptoethanol, calling for a better 3D immunostaining elution method. Alternatively, hyperspectral imaging[46], nonlinear optics[47], time-resolved fluorescence techniques[48],

and same-species antibody multiplexing[49] could be explored to extend the multiplexing capabilities of INSIHGT. Finally, although theoretically applicable, we have yet to apply the INSIHGT-based multi-round staining in tissues from other species.

Our discovery of boron clusters' capabilities to solubilize proteins globally in a titratable manner, combined with their bio-orthogonal removal with supramolecular click chemistry, can reach beyond

**Fig. 8 | INSIHGT enables non-destructive characterization and analysis of human clinical samples. a** A 3.5mm-thick human cerebellum triplex-stained for glial filaments (GFAP), neurofilament (NF-H) and blood vessels (*G. simplicifolia* lectin I). Orientation and coherence (or fractional anisotropy) visualization of neurofilament (**b**, NF-H) and glial filament (**c**, GFAP) via structure tensor analysis. **d** Enlarged view of the boxed area in **a** via post-hoc confocal microscopy. **e** Prototypical morphology of human cerebellar neurofilament inclusions, where their extensions may loop back to the Purkinje layer and occasionally to another inclusion body (lowest image). **f** Overview of the 1078 manually traced neurofilament inclusions across the cerebellar sample, color-coded by z-depth. Traditional 2D histology with special stains on pre-INSIHGT and post-INSIHGT processed samples. **g** H&E staining of human brain, **h** Left to right: Periodic acid-Schiff (PAS), Alcian blue, and Masson's trichrome staining of human kidney, mouse colon, and mouse kidney sections respectively. **i** The Next-Generation Histopathology Pathway. INSIHGT is compatible with traditional histological pipelines, empowering a multi-pronged approach to maximizing the information extracted from clinical samples.

histology applications. Given the surprisingly robust performance of INSIHGT in complex tissue environments, we envision they can be applied in simpler in vitro settings to control intermolecular interactions −particularly when involving proteins−in a spatiotemporally precise manner.

## Methods

### Ethical statement

For animal tissues, all experimental procedures were approved by the Animal Research Ethics Committee of the Chinese University of Hong Kong (CUHK) and were performed in accordance with the Guide for the Care and Use of Laboratory Animals (AEEC number 20-287-MIS). The housing of animals was provided by the Laboratory Animal Service Center of CUHK. For human tissues donated post-mortem, prior ethics approvals have been obtained and approved by the Joint Chinese University of Hong Kong-New Territories East Cluster Clinical Research Ethics Committee (approval number 2022.137), with consent obtained from the donor and his family.

### Chemicals and reagents

The antibodies utilized in this study were listed in Supplementary Table 2. All protein-conjugating fluorophores tested and their compatibility with INSIHGT were listed in Supplementary Table 3. Secondary Fab fragments or nanobodies were acquired from Jackson ImmunoResearch or Synaptic Systems, and all lectins were sourced from VectorLabs. Conjugation of secondary antibodies and lectins with fluorophores was achieved through *N*-hydroxysuccinimidyl (NHS) chemistry. The process was conducted at room temperature for a duration exceeding 16 h at antibody concentrations >3 mg/ml, using a tenfold molar excess of the reactive dye-NHS ester. Dodecahydro-*closo*-dodecaborate salts and other boron cluster compounds were procured from Katchem, while cyclodextrin derivatives were obtained from Cyclolab, Cyclodextrin Shop, or Sigma Aldrich.

We noticed occasionally the chemicals involved in the INSIHGT process require purification. Specifically, for $Na_2[B_{12}H_{12}]$, if insoluble flakes were noticed after dissolution in PBS, the solution was then acidified to pH 1 with concentrated hydrochloric acid, extracted with diethyl ether (Sigma Aldrich), and the organic solvent was removed and distilled off with a warm water bath. The residual $H_2B_{12}H_{12}$ was then dissolved in minimal amount of water, and neutralized with 1 M $Na_2CO_3$ solution until pH 7 is reached with no further evanescence. The solution was then concentrated by distillation under vacuum and dried in an oven.

For 2-hydroxypropyl-γ-cyclodextrin and sulfobutylether-β-cyclodextrin, if insoluble specks or dusts were noticed after dissolution in PBS, the solution was vacuum filtered through 0.22 μm hydrophilic cellulose membrane filters (GSWP14250) using a Buchner funnel before use. A slight brownish-yellow discoloration of the resulting solution would not interfere with the INSIHGT results.

For benzyl benzoate, if the solution is yellowish (possibly due to the impurities of fluorenone present), the solvent is poured into a metal bowl or glass crystallization dish and refrigerated to 4 °C until crystallization begins. If no crystallization occurs, a small crystal seed of benzyl benzoate obtained by freezing the solvent at −20 °C in a microcentrifuge tube can be put into the cooled solvent to kick-start the process. The crystals were then collected by vacuum filtration with air continuously drawn at room temperature until the crystals are white, which were warmed to 37 °C to result in clear, colorless benzyl benzoate. If the resulting colorless benzyl benzoate is cloudy, 3 Å molecular sieves were added to the solvent to absorb the admixed water from condensation, before filtering off to result in a clear colorless benzyl benzoate. This purified benzyl benzoate is ready to constitute BABB clearing solution for imaging.

### Human and animal tissues

Adult male C57BL/6 were utilized. These mice were housed in a controlled environment (22–23 °C) with a 12-h light-dark cycle, provided by the Laboratory Animal Service Center of CUHK. Unrestricted access to a standard mouse diet and water was ensured, and the environment was maintained at <70% relative humidity. Tissues were perfusion formaldehyde-fixed and collected by post-mortem dissection. In the case of immunostaining for neurotransmitters where Immusmol antibodies were used, the tissues were perfusion-fixed with the STAIN-perfect™ immunostaining kit A (Immusmol) with the antibody staining steps replaced with those in our INSIHGT method.

For human tissues, brain and kidney tissues donated post-mortem by a patient (aged 77 at the time of passing) were used in this study. Prior ethics approvals have been obtained and approved by the Joint Chinese University of Hong Kong-New Territories East Cluster Clinical Research Ethics Committee (approval number 2022.137), with consent from the donor and his family. Human dissection was performed by an anatomist (HML) after perfusion fixation with 4% paraformaldehyde via the femoral artery. The post-mortem delay to fixation and tissue harvesting was 4 weeks at −18 °C refrigeration, and the fixation duration was 1 week at room temperature. The corresponding organs were then harvested and stored in 1x PBS at room temperature until use.

### Screening deep staining approaches with in situ antibody recovery

4% PFA-fixed, 1mm-thick mouse cerebellum slices, 0.5 μg anti-parvalbumin antibody (Invitrogen, PA1-933), and 0.5 μg AlexaFluor 647-labeled Fab fragments of Donkey anti-Rabbit antibody (Jackson Immunoresearch 711-607-003) were used in this experiment to develop our method. Co-incubation of the secondary Fab fragment and primary antibody was utilized for 1-step immunostaining. All stainings were performed with an overnight immunostaining first stage at room temperature (unless specified otherwise) in various buffers, with subsequent recovery secondary stage at room temperature (unless specified otherwise) in various buffers, as detailed for each strategy below. The tissues were then washed in 1x PBSN, dehydrated with graded methanol, and cleared in BABB, before proceeding to imaging with confocal microscopy.

For the SDS/αCD system, immunostaining was performed in a solution consisting of 10 mM sodium dodecylsulphate (SDS) in 1xPBS, while recovery was performed with a solution consisting of 10 mM αCD in 1x PBS.

For the GnCl/GroEL+GroES system, immunostaining was performed in solution consisting of 6 M guanidinium chloride in 1x PBS, while recovery was performed with GroEL+GroES refolding buffer, consisting of 0.5 μM GroEL (MCLabs GEL-100), 1 μM GroES (MCLabs

GES-100), 2.5 mM adenosine triphosphate, 20 mM Tris base, 300 mM NaCl, 10 mM MgSO$_4$, 10 mM KCl, 1 mM tris(2-carboxylethyl)phosphine hydrochloride, 10% glycerol, with pH adjusted to 7.9[50].

For the sodium deoxycholate (SDC)/βCD system, immunostaining was performed in a solution consisting of 15 mM sodium deoxycholate (SDC) with 240 mM Tris base, 360 mM CAPS (*N*-cyclohexyl-3-amino-propanesulfonic acid), with pH adjusted to 8, while recovery was performed with a solution consisting of 15 mM βCD with 240 mM Tris base, 360 mM CAPS, with pH adjusted to 8.

For the Na$_2$[B$_{12}$H$_{12}$]/γCD system, immunostaining was performed in a solution consisting of 0.1 M Na$_2$[B$_{12}$H$_{12}$] in 1x PBS, while recovery was performed in a solution consisting of 0.1 M γCD in 1x PBS.

## Benchmarking experiments

We designed a stringent benchmarking scheme for quantitative evaluation of antibody penetration depth and signal homogeneity across depth for comparison across existing deep immunostaining methods, based on our previously described principles (Supplementary Fig. 1a)[6] The benchmarking experiment is carried out in two parts, the first part using a whole mouse hemisphere stained in bulk with anti-Parvalbumin (PV) antibodies with excess AlexaFluor 647-conjugated secondary Fab fragments—termed bulk-staining—after which the tissue is cut coronally at defined locations using a brain matrix and re-stained with anti-PV antibodies and AlexaFluor 488-conjugated secondary Fab fragments—termed cut-staining (Supplementary Fig. 1a). Hence, signals from bulk-staining can be distinguished easily from cut-staining and reveal different penetration depths of the two-staged immunostaining. We tested different deep immunostaining methods in the bulk-staining stage of the experiments, while the cut-staining was performed in 1× PBS with 0.1% Tween-20 as a conventional immunostaining buffer. The bulk-staining duration for INSIHGT was 24 h in benchmarking.

All benchmarking samples were perfusion-fixed with 4% paraformaldehyde (PFA) in 1× PBS followed by post-fixation in 4% PFA overnight at 4 °C, except for SHIELD and mELAST samples where the SHIELD protocol was used. In addition, the final RI matching where the benzyl alcohol/benzyl benzoate (BABB) clearing method was universally employed to standardize the changes in tissue volumes and hence penetration distance adjustments. The standardized optical clearing avoids the variability in fluorescent quenching and tissue shrinkage/expansion introduced by different RI matching agents. For bulk-staining during our benchmarking experiment, we followed the published protocols except for eFLASH and mELAST due to the lack of specialized in-house equipment.

For eFLASH[12], we stained the SHIELDed and SDS-delipidated tissue in the alkaline sodium deoxycholate buffer (240 mM Tris, 160 mM CAPS, 20% w/v D-sorbitol, 0.9% w/v sodium deoxycholate) and titrated-in acid-adjusting booster buffer (20% w/v D-sorbitol and 60 mM boric acid) hourly over 24 h to achieve a −0.1 ± 0.1 pH/h adjustment rate, using primary IgGs with secondary fluorophore-labeled Fab fragments. The tissue was then washed with 1× PBSTN (1× PBS, 1% v/v Triton X-100, and 0.02% w/v NaN$_3$) two times 3 h each before imaging.

For mELAST[7,13,14], we stained the SHIELDed and SDS-delipidated tissue with the antibody and Fab fragments in 0.2 × PBSNaCh (0.2× PBS, 5% w/v NaCh and 0.02% w/v NaN$_3$, 5% v/v normal donkey serum) first for 1 day at 37 °C without embedding the SHIELDed tissue in elastic gel nor compression/stretching, followed by adding Triton X-100 to a final concentration of ~5% and incubated for 1 more day. The tissue was then washed with 1× PBSTN 2 times 3 h each before imaging.

For CUBIC HistoVision[8] and iDISCO[17], the tissue was processed and stained as previously described[9]. The staining durations were 14 days for CUBIC HistoVision and 7 days for iDISCO (both using primary IgGs with secondary fluorophore-labeled Fab fragments).

For SHANEL[51], the tissue was first delipidated with CHAPS/NMDEA solution (10% w/v CHAPS detergent and 25% w/v *N*-methyldiethanolamine in water) for 1 week, then further delipidated with dichloromethane/methanol as in iDISCO, then treated with 0.5 M acetic acid for 2 days, washed in water for 6 h repeated 2 times, and then treated with guanidinium solution (PBS with 4 M guanidinium chloride, 0.05 sodium acetate, 2% w/v Triton X-100) for 2 days, blocked in blocking buffer (1× PBS, 0.2% v/v Triton X-100, 10% v/v DMSO, 10% goat serum) for 1 day, and finally stained in antibody incubation buffer (1× PBS, 0.2% v/v Tween-20, 3% v/v DMSO, 3% v/v goat serum, 10 mg/L heparin sodium) using primary IgGs with secondary fluorophore-labeled Fab fragments for 7 days.

For quantification, PV-positive cells were identified using a Laplacian of Gaussian filter, followed by intensity-based segmentation. These segmented masks allow the quantification of bulk- and cut-staining channel intensities, in addition to the distance transformation intensity, performed in MATLAB R2023a (MathWorks, US). For an ideal deep immunostaining, the bulk-immunostaining signals should be independent of the bulk-staining penetration distances computed with distance transform of the segmented tissue boundaries, and perfectly correlate with that of cut-immunostaining. This is often not the case, as "rimming" of bulk-staining signals inevitably occurs as a "shell" around the tissue due to more easily accessible antigens on the bulk-staining tissue surface. The rimming effect can be quantified by fitting a single-term exponential decay curve

$$\frac{bulk - staining\ intensity}{cut - staining\ intensity} = e^{-\tau(bulk-staining\ penetration\ distance)} \tag{1}$$

and evaluating the decay constant, tau (τ), across penetration depths, with τ → 0$^+$ as we approach the ideal case.

## Screening chemicals for INSIHGT

We first pre-screened the WCS by immunostaining for parvalbumin in 1 mm$^3$ of mouse cortex tissue cubes in the presence of WCS at 0.1 M, after 1 day of incubation at room temperature the staining solution was aspirated and 0.1 M corresponding cyclodextrin was added and incubated overnight. The tissue was then washed in PBSN for 15 min two times and cleared with the BABB method, and imaged. This procedure eliminated [B$_{12}$Br$_{12}$]$^{2-}$, [B$_{12}$I$_{12}$]$^{2-}$, and [PW$_{12}$O$_{40}$]$^{3-}$ (as cesium or sodium salts) as they do not give the correct immunostaining pattern or lead to tissue destruction. We tested [Fe(C$_5$H$_5$)$_2$]$^+$ (as the hexafluorophosphate salt) for the sake of completion as a low-charge large-sized cation.

To benchmark the ability in achieving deep and homogeneous immunostaining, the above benchmarking procedure was used. Mouse hemibrains were fixed, washed, and stained with 1 μg rabbit anti-parvalbumin antibody with 1 μg AlexaFluor 647-labeled donkey anti-rabbit secondary antibody Fab fragments in 0.1 M of the WCS. The staining proceeded for 1 day after which the solution was replaced with 0.1 M corresponding cyclodextrin (or its derivatives) and incubated overnight. The hemibrains were then washed in PBSN for 1 h two times, cut in the middle coronally and re-stained for parvalbumin using AlexaFluor 488-labeled secondary Fab fragments. The tissue was then washed, cleared with the BABB method, and imaged on the cut face using a confocal microscope.

## INSIHGT

A detailed step-by-step protocol used in this study has been given below. As a general overview, tissues were typically fixed using formalin or 4% paraformaldehyde, thoroughly washed in PBSN, and pre-incubated overnight at 37 °C in INSIHGT buffer A. The tissues were then stained with a solution containing the desired antibodies, Fab fragments, lectins, and SBEβCD-complexed nucleic acid probes in INSIHGT buffer A, ensuring a final [B$_{12}$H$_{12}$]$^{2-}$ concentration of 0.25 M. Staining duration varied from 6 h to 10 days based on tissue size, antigen, and required homogeneity (please see the calculation of time *t* in the step-by-step protocol). Post-staining, the solution was aspirated and replaced with INSIHGT buffer B (0.25 M 2-hydroxypropyl-γ-

cyclodextrin in PBS) without prior washing, followed by a minimum 6-h incubation with adequate shaking of the viscous buffer. After sufficient PBSN washing, tissues were ready for imaging or clearing. Over incubation for any steps up to 60 days was tolerable. After imaging, the antibodies can be eluted with 0.1 M sodium sulphite in INSIHGT buffer A at 37 °C overnight.

## Screening antibodies compatible with INSIHGT

To test antibodies in a high-throughput manner, we compiled a list of antibodies, reviewed their tissue expression and staining patterns in the literature, and then obtained the respective tissues known to have positive staining. These tissue blocks or entire organs were then washed, dehydrated, delipidated, rehydrated, washed, and infiltrated with INSIHGT solution A as described in the INSIHGT protocol. These INSIHGT-infiltrated tissues were then cut into ~1 mm³ tissue cubes and placed in a 96-well plate as indicated in the list, with each well containing 70 µl of 1x INSIHGT solution A. About 0.5 µg of the primary antibody to be tested was then added and 0.5 µg of the corresponding AlexaFluor 647 or AlexaFluor 594-conjugated secondary antibody Fab fragment. The AlexaFluor 647 and 594 fluorophores were chosen for to minimize interference from any tissue autofluorescence on the result interpretation. For a total volume and antibodies added two each well, an equal volume of 2x INSIHGT solution A was then added to ensure the final concentration of 1x INSIHGT solution A. The plate was then sealed and the staining was allowed to proceed in the dark overnight at room temperature. The tissues were then washed in INSIHGT solution B for 2 h, PBSN for 1 h for two times, and then dehydrated with through 15 min-incubation of 50% methanol, 100% methanol, and 100% methanol. The tissues were then cleared in BABB for 15 min and proceeded to imaging. The total fixed tissue-to-image time for the antibody compatibility test is <36 h.

## Comparison between 2D histological staining of post-INSIHGT and control tissues

Mouse and human samples were pre-processed as described above. Tissues were divided into the post-INSIHGT treated group which underwent the INSIHGT protocol with 3 days of INSIHGT A incubation without the application of antibodies and 6 h of INSIHGT B incubation, plus BABB clearing, and the control group which was immersed in PBSN for an equivalent period of time. Both groups were immersed in 70% ethanol, preceded by the immersion in 100% ethanol for the post-INSIHGT group (which were in BABB), and in 50% ethanol for the control group (which were in PBSN). Tissues were then immersed in 100% ethanol, xylene, and paraffin as in the standard paraffin embedding process. The embedded tissues were cut into 5 µm (human) or 10 µm (mouse) sections followed 2D histological staining with special stains. Following standard protocols, H&E staining was performed on human brain and kidney, PAS staining was performed on human kidney, Alcian blue staining was performed on mouse colon, and Masson trichrome staining was performed on mouse kidney samples.

## Microscopy

Confocal microscopy was performed using a Leica SP8 confocal microscope equipped with excitation lasers at 405 nm, 488 nm, 514 nm, 561 nm, 649 nm, with detection using a 10× (NA 0.4, Leica HC PL APO ×10/0.40 CS2) or a 40× oil-immersion (NA 1.30, Leica HC PL APO 40×/1.30 Oil CS2) objective and a tunable emission filter. A custom-built MesoSPIM v5.1[52] was used for light-sheet microscopy equipped with lasers at 405 nm, 488 nm, 514 nm, 561 nm, 633 nm, and 675 nm, with detection using an Olympus MVX-ZB10 zoom body with a magnification range from 0.63×–6.3×. The equipped emission filters were from AHF, including QuadLine Rejectionband ZET405/488/561/ 640, 440/50 ET Bandpass, 509/22 Brightline HC, 515/LP Brightline HC Longpass Filter, 542/27 BrightLine HC, 585/40 ET Bandpass, 594 LP Edge Basic Longpass Filter, 660/13 BrightLine HC, 633 LP Edge Basic Longpass Filter, and a 685/LP BrightLine HC Longpass Filter. Two-photon tomography was performed at 780 nm excitation[9] using a 16× objective (NA 0.8, Nikon CFI75 LWD 16X W), equipped with four emission filters (ThorLabs 460-60, Semrock 525/50, Semrock 607/70, and Chroma ET 670/50). Basic image acquisition parameters for all microscopy images in this study were listed in Supplementary Table 4.

## RNA and DNA quality control

Control and INSIHGT-treated samples following the 1 mm³ treatment timeline were re-embedded in paraffin wax and sent for nucleic acid integrity, sequencing, and bioinformatics analysis services provided by the BGI Hongkong Tech Solution NGS Lab. RNA integrity number analysis was performed using the Qubit Fluorometer. Whole genome DNA quality analysis was performed using the Agilent 2100 Bioanalyzer system. Sequencing was performed using the DNBSEQ™ sequencing technology platform.

For transcriptomic comparison, the total clean bases were 11.2 Gb and 10.97 Gb for the control and INSIHGT-treated samples, respectively. The clean reads ratio after filtering was 90.64% and 89.96%, respectively. For whole genome sequencing, The total clean bases were 114.5 Gb and 125.2 Gb for the control and INSIHGT-treated samples, respectively, with both samples having a clean data rate of 100% and a mapping rate of 99.96%.

## RNA FISH HCR with INSIHGT

Our RNA FISH HCR protocol is largely adapted from Choi et al.[53]. The post-INSIHGT samples were first fixed in 4% PFA for 1 day. The samples were then pre-incubated in pre-hybridization buffer until the tissue sank to the bottom, and hybridized in hybridization buffer at 37 °C overnight. The next day, the tissue was washed in probe wash buffer for 1 h two times at room temperature, pre-incubated in amplification buffer for 30 min, followed by HCR amplification by incubating in amplification buffer with the addition of 30 pmol of fluorescently-labeled HCR hairpins and incubated overnight at RT. Note that the HCR hairpins were snap-cooled (heated at 95 °C for 2 min and cooled to RT for 30 min) in 10 µL 5× SSC buffer before application to ensure hairpin structures are formed[54]. The samples were then washed thoroughly in 500 µL probe wash buffer for 30 min × 3 times to mitigate non-specific binding and later subjected to confocal imaging.

The HCR probes which hybridize on the mRNA targets were custom-designed following the approach by Choi et al.[53], as shown in Table 1, and were purchased from Integrated DNA Technologies.

## Image processing

No penetration-related attenuation intensity adjustments were performed for all displayed images except for the 3D renderings (but not 2D cross-sectional views) in Fig. 3 and Supplementary Movie 1 to provide the best visualization of an internal signal. For samples imaged with two-photon tomography, we noticed a thin rim attributed to the heat produced during the gelatin embedding process (which we verified by repeating the staining and confirming its absence with light sheet microscopy). We employed an intensity transformation mask based on the exponent of the distance from the whole organ mask surface. Image segmentation was performed with Cellpose 2.0[28] for cells implemented in MATLAB R2023b or Python, or with simple intensity thresholding. Affine and non-linear image registration was performed in MATLAB R2023a or manually in Adobe After Effects 2020 using the mesh warp effect and time remapping for z-plane adjustment. Image stitching was performed either with ImageJ BigStitcher plugin[55] or assisted manually with Adobe After Effects 2020 followed by tile allocation using custom-written scripts in MATLAB R2023a.

3D image visualization and Movie rendering were performed with Bitplane Imaris 9.1, which were done as raw data with brightness and contrast adjustments, except for the whole mouse brain imaged with

two-photon tomography. To remove their slicing artifacts, we resliced the volume into x-z slices, performed a z-direction Gaussian blur, followed by a 2D Fourier transform and filtered out non-central frequency peaks before inverting the transform. Finally, a Richardson-Lucy deconvolution was performed with a point-spread function elongated in the x-z direction, and resliced back into x-y slices.

## Segmentation and analysis of podocyte-to-PEC microfilaments in mouse kidneys

Podocyte-to-PEC microfilaments of 14 mouse kidneys were manually traced via the SNT plugin in ImageJ[56]. Path properties of the tracings were then exported for further analysis using custom codes in MATLAB R2023a. Distance transforms were performed under manually curated glomerulus and Bowman space masks, such that each voxel value corresponds to the distance between that voxel and the nearest nonzero voxel of the Bowman space mask. Path displacement $d_{fil}$ was computed via Pythagoras theorem using the start and end coordinates of the filament. Minimal distance $d_{min}$ is defined as the voxel value difference between the start and end coordinates. Path length $d_{path}$ is directly measured via SNT. Tortuosity is defined as $d_{path}/d_{fil}$, skewness is defined as $d_{fil}/d_{min}$, and the angle of take-off is defined as the angle between the unit gradient vector of the distance transform and the unit path displacement vector. The geodesic distance $d_A(p,q)$ between voxels $p, q \in A$ is defined as the minimal of length $L$ of path(s) $P = (p_1, p_2, \ldots, p_l)$ connecting $p, q$, where $A$ is the set of all voxels constituting the surface of the glomerular mask[57]:

$$d_A(p,q) = \min\{L(P) : p_1 = p, p_l = q, P \subseteq A\} \quad (2)$$

Correlation statistics were then performed via GraphPad Prism version 8 for Windows, GraphPad Software, Boston, Massachusetts USA, www.graphpad.com. Tracing and statistical analysis for the human cerebellar neurofilament inclusions were performed analogously.

## Spatial orientation and fractional anisotropy visualization of human cerebellum neural and glial filaments

To visualize cerebellar neural and glial fibers in their preferred orientations, we performed structure tensor analysis with orientation-based color-coding in 3D.

In detail, let $G : \mathbb{R}^3 \times \mathbb{R}_+ \to \mathbb{R}$ be a 3D Gaussian kernel with standard deviation $\sigma$:

$$G(x,y,z,\sigma) = \frac{1}{(2\pi\sigma^2)^{\frac{3}{2}}} \exp\left(-\frac{x^2 + y^2 + z^2}{2\sigma^2}\right) \quad (3)$$

Define a 3D image as a function $I : \mathbb{R}^3 \to \mathbb{R}$ which outputs the spatial voxel values. The gradient $\nabla I : \mathbb{R}^3 \to \mathbb{R}^3$ of $I$ at each voxel is obtained by convolving $I$ with the spatial derivatives of $G$:

$$\nabla G(x,y,z,\sigma) = \left(\frac{\partial G}{\partial x}, \frac{\partial G}{\partial y}, \frac{\partial G}{\partial z}\right) \quad (4)$$

$$\nabla I = I * \nabla G \quad (5)$$

where * denotes the convolution operation.

Compute the structure tensor $T : \mathbb{R}^3 \to \mathbb{R}^{3\times3}$ as the outer product of $\nabla I$ with itself:

$$\mathbf{T}(x,y,z) = \nabla \mathbf{I} \otimes \nabla \mathbf{I} \quad (6)$$

$\mathbf{T}$ is then smoothed over a neighborhood $N$ via convolution with $G$ to give $\bar{\mathbf{T}}$:

$$\bar{\mathbf{T}}(x,y,z) = G * \mathbf{T}(x,y,z) \quad (7)$$

$$\bar{\mathbf{T}}(x,y,z) = \begin{bmatrix} \langle I_x^2 \rangle_N & \langle I_x I_y \rangle_N & \langle I_x I_z \rangle_N \\ \langle I_y I_x \rangle_N & \langle I_y^2 \rangle_N & \langle I_y I_z \rangle_N \\ \langle I_z I_x \rangle_N & \langle I_z I_y \rangle_N & \langle I_z^2 \rangle_N \end{bmatrix} \quad (8)$$

where $\langle \cdot \rangle_N$ represents the Gaussian-weighted smoothing over $N$[58,59].

Eigendecomposition of $\bar{\mathbf{T}}$ is then performed to define the shape (eigenvalues, $\lambda$) and the orientation (eigenvectors, $\mathbf{v_e}$) of the diffusion ellipsoid. The fractional anisotropy ($FA$) is then computed from $\lambda$:

$$FA = \sqrt{\frac{(\lambda_1 - \lambda_2)^2 + (\lambda_2 - \lambda_3)^2 + (\lambda_3 - \lambda_1)^2}{2\left(\lambda_1^2 + \lambda_2^2 + \lambda_3^2\right)}} \quad (9)$$

where $FA$ ranges from 0 (complete isotropic diffusion) to 1 (complete anisotropic diffusion)[60].

The tertiary (least) eigenvalue-associated eigenvectors were then extracted for the 3-dimensional image volume, with the 4th dimension encoding the corresponding vector basis magnitudes. To visualize the orientation of fibers in the context of the image, the eigenvectors were intensity-modulated with both the fractional anisotropy and the original image voxel values, and represented as a 3D RGB stack for visualization in Imaris.

## Multi-round multiplexed 3D image processing and analysis

As the images were acquired across multiple rounds on a confocal microscope, we encountered the issues of misalignment and z-step glitching due to piezoelectric motor errors. Hence, the tiles of images can neither be directly stitched nor registered across multiple rounds. A custom MATLAB code was written to manually remove all the z-step glitching, followed by matching the z-steps across multiple rounds aiding by using the time-remapping function in Adobe After Effects, with linear interpolation for the transformed z-substacks. The resulting glitch-removed, z-matched tiles were then rigid registered using the image registration application in MATLAB, followed by non-rigid registration for local matching. Finally, the registered tiles were stitched for downstream processing.

Before segmentation, all non-vessel channels underwent background subtraction. They were then summed to capture the full morphology of stained cells, followed by segmentation using Cellpose 2.0[28]. A custom model was trained and used based on 2D excerpts of the images until adequate segmentation accuracy was achieved by manual inspection. The final test image segmentation has a Dice Coefficient (or F1-score) of $0.9354 \pm 0.0596$ and Jaccard Index of $0.8824 \pm 0.1023$, provided as mean $\pm$ S.D. on six excerpted test images. Vessels were segmented based on their staining intensity, and a distance transform was used to obtain the distance from vessels for all voxels. The cell masks subsequently facilitated the acquisition of the statistics for all stained channels.

UMAP was performed in MATLAB R2023a using the UMAP 4.4[55,61] package in a nested manner, incorporating the means and standard deviations of all immunostaining intensities, as well as the distance to the nearest blood vessel. An initial UMAP (with "min_dist" = 0.05, "metric" = "euclidean", and "n_neighbors" = 15) was applied to each image stack tile, followed by DBSCAN clustering (using the default value $\varepsilon = 0.6$) to eliminate the largest cluster based on cell count. The remaining cells were subjected to a second UMAP (with the same parameters), where another round of DBSCAN clustering (with the same parameters) yielded the final cell clusters for analysis. The choice of UMAP parameters was based on an online guide (https://umap-learn.readthedocs.io/en/latest/api.html) and visual inspection on the reasonable clustering results.

Violin plots for each clustered cell type's distance from neuropeptide Y-positive fibers were obtained by creating a distance transformation field from the segmented fibers. Segmented cell masks were used to compute the mean intensity value of the distance transformation field. The pairwise distances of the clustered cell types were obtained for the 30 nearest neighbors, followed by calculating the mean and SD for the coefficient of variation. The gramm package in MATLAB R2023a was used for plotting some of the graphs[62].

## Statistics and reproducibility

For Fig. 2c, Supplementary Figs. 6, 7, one-component exponential regression was applied for curve fitting, and Pearson's correlation coefficient was computed for the scattered plot in Fig. 2d. Two-sample unpaired *t*-test was employed for Supp. Fig. 28 The staining and imaging experiments in Fig. 2–8, were repeated with at least two independent samples in the same or similar condition with slight modifications, such as using similarly sized tissues of similar characteristics (especially for human samples), using different staining antibodies and marker choice, or staining durations. All the results were reliably reproduced in accordance with the expected outcome of the methods. No method was used to predetermine sample size.

## Reporting summary

Further information on research design is available in the Nature Portfolio Reporting Summary linked to this article.

## Data availability

The raw imaging data in this paper are too large for public deposit. They will be made available upon request to the corresponding author (H.M.L). The benchmarking experiment dataset has been deposited and made available for analysis at Code Ocean (capsule link: https://doi.org/10.24433/CO.4249201.v1). The data associated with Fig. 7f, g were provided in the Source Data File. Source data are provided with this paper.

## Code availability

The code for benchmarking experiment analysis along with sample data has been deposited and made available at Code Ocean (capsule link: https://doi.org/10.24433/CO.4249201.v1).

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

## Acknowledgements

We would like to express our deepest gratitude to the tissue donor and his family for their generosity and wisdom in supporting scientific studies. We thank William Wu for access to confocal microscopy; Cathy Shuk Ling Chan for manual data annotation and analysis; Ka Wai Chan for administrative support to the project. Figures 1and 6h were partly created with BioRender.com. The project was supported by the Midstream Research Program for Universities (MRP/048/20, H.M.L.) of the Innovation and Technology Council of Hong Kong, a Direct Grant for Research 2022/23 (2022.072, H.M.L.) of the Chinese University of Hong Kong, and the Chinese University of Hong Kong Research Committee Group Research Scheme (GRS) 2021–22 (granted to Y.K.W.).

## Author contributions

Conceptualization: H.M.L. Methodology: C.N.Y., J.T.S.H., R.A.A.C., T.C.Y.W., B.T.Y.W., N.K.N.C., L.Z., E.P.L.T., H.M.L. Investigation: C.N.Y., J.T.S.H., R.A.A.C., T.C.Y.W., B.H., B.T.Y.W., N.K.N.C., L.Z., E.P.L.T., Y.T., J.J.X.L., Y.K.W., H.M.L. Visualization: C.N.Y., J.T.S.H., R.A.A.C., H.M.L. Funding acquisition: Y.K.W., H.M.L. Project administration: H.M.L. Supervision: H.M.L. Writing: C.N.Y., H.M.L.

## Competing interests

C.U.H.K. filed a patent application in part based on the invention described in this paper with H.M.L. and C.N.Y. as the inventors. The associated patent, owned by C.U.H.K., was exclusively licensed to Illumos Limited, of which H.M.L. is a co-founder. The remaining authors declare no competing interests.
