## [Transparent Peer Review file · Nature Communications]

INSIHGT: An accessible multi-scale, multi-modal 3D spatial biology platform

Corresponding Author: Dr Hei Ming Lai

Version 0:

Reviewer comments:

Reviewer #1

(Remarks to the Author)

In this manuscript, the authors presented an elegant and practical concept to achieve homogenous deep 3D histochemistry termed INSIGHT. The authors achieved increased tissue penetration and demonstrated the labeling of macromolecules in tissues down to centimeter depth using the weakly coordinating superchaotropes, and reinstated the antibody-antigen binding using macrocyclic compounds for in situ host-guest reaction. The authors benchmarked their technique with existing deep 3D immunohistochemistry approaches using a rigorous approach, e.g., bulk-staining vs. cut-staining, showcased the superior performance of INSIGHT, and evaluated an impressive list of over 300 antibodies for INSIGHT. The authors demonstrated the versatility of their approach to different biological targets, including small molecules, proteins, lectins, and nucleic acids, various tissues, volumes, and origins. They further showed how the method can be adapted to clinical samples without apparent tissue destruction. The creative application of a simple concept to achieve fast and deep histochemistry is impressive. Overall, the innovative study is well-designed and technically sound, with carefully designed control experiments. The claim is supported by the data, and the protocol could benefit the broad community of 3D histopathology and spatial biology. As such, the manuscript is suitable for publication in Nature Communications after addressing the following comments.

Major comments:

1) The concept of using weakly coordinating superchaotropes to modulate the antibody solubility and tissue penetration depth is interesting. How was the working $\text{Na}_2[\text{B}_{12}\text{H}_{12}]$ concentration obtained? What is the reason for the different concentrations used during screening and INSIGHT protocol (0.1 M vs. 0.25 M)? Have the authors systematically evaluated how the concentration affects the superchaotropic effect on the targets? Would increasing or decreasing concentrations increase/decrease the penetration depth?

2) In the manuscript, the authors mentioned that IgG needs two-step staining, as in primary and secondary staining. In contrast, Fab can be done in a one-step manner by co-incubating the primary and secondary antibodies with the sample. What was the primary reason? Additionally, it would help if the authors could add a discussion on F(ab)'_2 . Can it be used in single-step staining or two-step staining is required?

3) It would be helpful to have a positive control to evaluate the labeling efficiency of INSIGHT. This can be achieved by using a protein target that expresses a GFP fusion protein and assessing anti-GFP immunostaining. For instance, using antibodies such as Abcam ab252881 or ab6673 (refer to Table S2), evaluate the number of cells exhibiting both GFP and Alexa Fluor 647 signals relative to the total number of GFP+ cells. Or at least some extended comments would be helpful, e.g., under the context where the tissue target may be low-abundant.

Minor comments:

1) The terms weakly coordinating ions, weakly coordinating anions, and weakly coordinating Superchaotropes are interchangeably used and could be clearer. For instance, "We first pre-screened the weakly coordinating anions by immunostaining for parvalbumin in 1 mm³ of mouse cortex tissue cubes in the presence of 0.1M weakly coordinating

superchaotropes, after 1 day of incubation at room temperature the staining solution was aspirated and 0.1M corresponding cyclodextrin was added and incubated overnight." It may be useful to have a sentence defining the connection between these terms.

2) In the section "In situ host-guest chemistry for three-dimensional histology," the author mentioned, "To quantitatively compare the signal, we segmented the labeled cells and compared the ratio between the deep immunolabeling signal and the reference signal against their penetration depths." It would be clearer for the readers if the terms "bulk staining" (deep immunolabeling) and "cut staining" (reference signal) were defined in the text or methods.

3) In the same section, the authors wrote, "This is evident as the cut-staining intensity profile of INSIGHT was similar to that of iDISCO (Fig. S7) which has identical tissue pre-processing steps." However, in Fig S7, the profiles were not that similar. Could the authors elaborate on the observation?

4) In Supp Fig 1. A, it would help to mention whether the shown images are bulk staining or cut staining channel in the figure caption. From the caption from Figure 2B, should it be cut staining channel? It would also help if the 'mol', 'pur' and 'gran' labels on the S1A were defined in the figure caption. Supp Fig 1. B, is the blue stain DAPI?

5) Supp Fig. 4 is somewhat difficult to interpret, especially the two columns representing shallow ($\frac{1}{3}$) and deep ($\frac{2}{3}$) imaging z-depths. Could the authors elaborate on it? Additionally, it is unclear what host/guest system is used in the example shown.

6) Figure 3A, the z-depth definition for color code appears missing.

7) Alexa Fluor 647 was selected to minimize the interference from tissue autofluorescence. Could the author comment on the use of other NIR dyes, such as Cy5 and ATTO 647N? Alexa Fluor 647 is known to photobleach quickly.

8) On page 39, the INSIGHT protocol lists the equations for the minimum staining time and post-incubation time. The equations are very useful for someone reproducing the results. Are equations empirically determined? For instance, if a tissue section is 10 mm thick, would the staining time be 72 h? Would these equations be generally applicable to a variety of tissues? Some comments would be helpful.

9) It's exciting that the authors observed membrane filaments in tissue (Fig 7). Elongated membrane fibers, such as tunneling nanotubes, have been reported by in vitro studies for intercellular material transfer in various cell type (PMID: 34795441, 25571977, 37202391). Some discussions of observation with existing studies would help enhance the significance of the observation.

(Remarks on code availability)

See the attachment "INSIGHT_code_review_NC.pdf" with screenshots.

Reviewer #2

(Remarks to the Author)

In this paper the authors developed a user-friendly 3D histochemistry method In situ Host-Guest Chemistry for Three-dimensional Histology (INSIGHT). This method has several advantages, including homogeneous probe penetration up to centimeter depths, quantitative and highly specific immunostaining signals, a fast and affordable workflow that can accommodate different tissue sizes and shapes, simple immersion-based staining at room temperature that can be scaled and automated, and direct application to unlabeled mouse and human tissues using off-the-shelf antibodies.

In this paper they performed a comparison between INSIGHT and the previous 3D technique such as (SHANEL, IDISCO, SHIELD, CUBIC, eFLASH, mELAST) to show the difference in antibodies and probe penetration in the samples. They evaluated 357 antibodies and discovered that 323 of them (90.5%) produced the desired immunostaining patterns when manually confirmed against the human protein atlas and/or current literature which is a high number of comparisons with the other techniques. They also analyzed INSIGHT-treated samples for RNA integrity number (RIN) and whole genome DNA extraction. They discovered that each phase of the INSIGHT methodology did not result in a substantial decrease in RNA integrity number (RIN). For that reason, this technique is particularly significant for the evaluation of the protein and RNA in the same sample.

Finally, they demonstrate the cost-effectiveness of INSIGHT, a three-dimensional (3D) spatial biology approach based on superchaotropes and host-guest chemistry. This allows for multi-modal readout of tissue biomolecules in biological systems at centimeter scales.

Here are some questions for the authors:

1. What is the magnification for 3D imaging?

2.

3. Can you provide the number of days needed for staining the full adult mouse brain for each approach compared to INSIGHT (for the experiments you conducted)? As indicated, the stages for iDisco are similar to those for INSIGHT, thus I'd want to know the duration of the staining step.

4. Please provide further information on the type of probe used, where it was ordered, and the time of incubation for detecting mRNA and DNA. The step of mRNA detection is not clear.

5. Are there any control trials for 2D mRNA detection to compare expression levels to INSIGHT?

6. Have you used mRNA detection for entire tissues, such as mouse brains or any large samples?

(Remarks on code availability)

The authors have clearly identified the limitations of current immunolabeling methods for obtaining accurate 3D quantification using light sheet microscopy.

The proposed method is undoubtedly of great interest to the scientific community and I've focused more specifically on the claim for homogenous probe penetration, in which I have a particular interest.

Although I was able to run the scripts that generate the figures from the analysis results provided, I was unable to reproduce these results from the data supplied.

In summary, I was unable to reproduce all the results due to:

1- missing data for alternative methods.

2- the information associated with the data is unclear. Data and associated masks are not provided in the same dimensions - channels are interleaved but not clearly labeled - code is very sparsely commented, which can quickly lead to processing errors.

3- resizing operations in benchmarking_analyze.m don't seem right to me and can introduce a significant bias in correlation analyses between the two channels.

4- cell identification and segmentation methods are correct, but extremely parameter-dependent. The slightest change in the input data and/or parameters can therefore dramatically alter the results. These parameters should be clearly indicated at the beginning of the script, along with all other parameters related to the analysis. This is particularly important when you need to compare several alternative methods, to make sure that the results presented depend on the specificities of the methods and not on the parameter settings.

More specific indications of the problems encountered are presented below, using as support the README file provided in code-ocean.

I am sure that optimized analysis code / data / documentation must be available and I remain at the authors' disposal to test them and make them easily testable by the scientific community.

Thank you for your work.

Reviewer #3

(Remarks to the Author)

The study proposes INSIHGT "In situ Host-Guest Chemistry for Three-dimensional Histology" a new method to perform homogeneous probe penetration on both mouse and human tissues, to perform specific and multiple markers immunostaining and RNA staining. The proposed protocol is simple and easy to implement in any lab and compatible with scaling and automated analysis.

The work is innovative and will help the scientific community towards the routinely application of 3D histology. However, I have some important points that need to be clarified in order to demonstrate the replicability of the proposed results.

- It is not clear from either the abstract, introduction, or results with which kind of clearing the INSIHGT approach was optimized, only by looking at the method is possible to understand it. Please add it when presenting the INSIHGT method and its identification.
- The authors said "Notably, after washing, only a negligible effect of [B12H12]₂-treatment will remain within the tissue.": to verify the integrity of the nanostructure of tissues after INSIHGT it is desirable to have some TEM or SEM images that will provide information about the ultrastructure of the tissue after the proposed protocol.
- The text in all the figures is too small and not visible, please increase the size of the text to make them visible.
- Please specify the microscope used for every image presented and the resolution used for the acquisitions so as to permit the correct evaluation of the results. Moreover, please give more details about the microscopes and the configuration used for the analysis. E.g: how many laser lines were present in each microscope, which objectives were used, etc.. All the information needed to repeat the experiment by other groups need to be provided. For example: for the 7 channels acquisition with confocal microscopy of 2mm depth samples there is no information about the lasers used, the objective, and therefore the resolution of the images.
- Please add some z-stack reconstruction to demonstrate the good penetration and contrast of the staining in the samples, 3D reconstructions are very nice to look at, but does not permit to evaluate correctly the staining capability.
- In figure 5 to demonstrate the penetration of the antibody inside the tissue the sample was cut and image with confocal microscopy, why the authors did not provide also a 3D reconstruction of the whole block with the mesospim? Even if the resolution is lower, it can detect the signal from the inner part of the sample proving the good antibody penetration. Then to visualize the sample at higher resolution it can be cut and image with confocal microscopy. I suggest the authors analyze an additional human brain sample with the dimension of ~2 x 2 x 3 cm³ (or the larger portion INSIHGT can analyze) and image it with the mesospim to prove the good penetration of the antibody deep inside the sample.
- For the 7 imaging rounds of 28 neuronal marker in 2 mm-thick hypothalamus slice, please provide images of all the separate channels (e.g for one plane of the sample) in order to provide to the reader a clearer picture of the data obtained with the analysis and the quality of the staining since merged images are very difficult to interpret. Please provide information of the resolution used for the acquisition and the time needed for the acquisition and the analysis in order to give information about the possibility of using the approach and the possible scaling up for routinely analysis.
- For the multiround analysis please provide the accuracy of cell segmentation obtained using Cellpose 2.0 and the quantity of GT that was performed for each analyzed channel in order to use the tool (or if the author used a pre-trained dataset). Moreover: cellpose is a software that was presented as a method to analyze image in 2D. Since the reconstruction was

performed in 3D how the authors used the software to obtain 3D analysis of the samples? Why they didn't used alternative methods developed for 3D analysis as StarDist or BCFind?

- The behavior of the DBSCAN clustering algorithm is influenced by two internal parameters: the search radius ϵ around separate data elements and the minimum number of data points within such a radius required for an element to be considered as belonging to a local "high-density" cluster. Together, these parameters define a minimum threshold on the local data density and thus affect the density and number of the resulting clusters. What are the values adopted by the authors, and what criteria drove the selection of such values? Did the authors evaluate the possible different outcomes of different (ϵ , N_{min}) combinations?

- In figure 6 for both the cell and NYP fiber segmentation please provide high resolution images of the segmentation obtained with the analysis. From the small images presented in figure 6 the quality of the cell segmentation is not appreciable, and the NYP segmentation is not provided at all. Please include information about the accuracy of the segmentation and the presence of false positive and negative recognitions.

- In the discussion the authors say: "Immunostaining penetration homogeneities for larger tissues and denser antigens can be further enhanced": to be more precise please provide the exact depth limit achievable with INSIGHT so that the reader can have a clear knowledge of the possibility of the method.

(Remarks on code availability)

I don't have the expertise to review the code.

Reviewer #4

(Remarks to the Author)

(Remarks on code availability)

Reviewer #5

(Remarks to the Author)

(Remarks on code availability)

Reviewer #6

(Remarks to the Author)

In this manuscript, the authors developed a two-step staining method, named INSIGHT, that enabled multi-modal analyses by simultaneously labeling a wide range of biomolecules in a three-dimensional tissue. The method reported in this well-written manuscript is a breakthrough in many biological fields. I hope that my comments below are helpful in improving the manuscript.

Comment #1

In the Abstract section, introducing INSIGHT in a more concrete way might be reader friendly. For example, adding "two-step staining (method)" and "simultaneously (visualized)" in the third sentence and "deep inside the tissue (up to centimeter scale)" in the fourth sentence might be informative.

Comment #2

According to the Microscopy subsection in the Methods section on the page 35, three types of microscopies (confocal, light-sheet and two-photon) were used. For clarity, please consider adding information on the type of microscopy used for image acquisition in each figure legends throughout the manuscript.

Comment #3

In the Figure 2C, the fluorescent "ratio" between bulk- and cut-stain was compared. Describing the fluorescence intensity in arbitrary unit, in addition to the ratio, for each staining method would enhance understanding of the experiment.

Comment #4

In the Figure 4B, not in the Figure 4C, the degree of lysine methylations was examined. Please correctly refer to this point in the main text (the second "Fig. 4B-C" should be corrected to "Fig. 4B").

Comment #5

Application of INSIHGT to nucleic acid probes (Figures 4E-G) is one of the key features. In the “Graphical summary of INSIHGT protocol” on the page 42, use of INSIHGT C solution was nicely summarized. Adding brief explanation sentences on the experimental procedure in the corresponding paragraph on the page 8 might be informative.

Comment #6

The present reviewer was enabled to find the pathological contribution of the subcerulean nuclei in the reference #25 while, according to Figure 5G and 5H, the sampled area is the locus ceruleus (LC). Please check the sampled area and add figure legends for Figures 5G-5I that are missing in the current manuscript. In addition, as is already provided in the Supplementary Figure 15, the full spellings of the brain areas were required in the figure legends of the main text: inferior colliculus (IC), red nucleus (RN), substantia nigra (SN), superior colliculus (SC).

Comment #7

The sentence (“Such a ratio-histopathology approach ... for predicting underlying neurodegeneration.”) might be an overspeculation on the application of INSIHGT. One of the breakthroughs of INSIHGT lies in bridging the microscopic and mesoscopic imaging modalities at a molecular level. Given the latter point, deleting the former sentence from the manuscript would enhance the scientific soundness without reducing the impact of INSIHGT.

Comment #8

Twenty-five molecular markers instead of 50 markers were examined in Supplementary Figure 18. Please revise the first sentence on the page 10 (“a subset of 84,139 cells based on 50 markers”).

Comment #9

Attempts to link cell morphology and single-cell transcriptomics have been made, for example in a paper (PMID: 35347329). Therefore, the expression on the page 10 (“which is inaccessible to spatial transcriptomics or single-cell multi-omics”) is too strong. Please consider reducing the tone here. Alternatively, the strength of INSIHGT is enabling mapping “proteins” of three-dimensional tissues at a subcellular level, which could be emphasized. Indeed, measuring the distance between NPY fibers and the nearest cell membrane is meaningful because INSIHGT was applied to the three-dimensional tissue.

Comment #10

In an animal model of crescentic glomerulonephritis, the link between the glomerulus tuft and Bowman’s capsule is reported (PMID: 11562404). This linking structure in the normal condition, however, is hardly documented to the best of the reviewer’s knowledge. Please consider reducing the tone in the sentence referencing # 27-31. Alternatively, the strength of INSIHGT here is that it carries the information on the constituent such as wheat germ agglutinin (WGA) of this linking structure. In addition, adding discussion on the reason INSIHGT enabled visualization of this structure might be informative.

Comment #11

The reviewer has two minor suggestions in the kidney subsection on the page 10. The inverse relationship reported in Figure 7G is difficult to understand at a glance that would be supported by Pearson coefficient. In addition, the word “overarching (PECs)” is confusing because the linking structure belongs to the podocyte. Please consider these points.

Comment #12

The human brain tissue derived from a single DLB patient according to page 31. Therefore, discussion on the pathology of the NF-H-intense inclusions, if applicable, might be informative.

Comment #13

Adding the reference numbers of approval for the use of human- and animal-derived tissue on pages 30 and 31 are encouraged.

Comment #14

Supplementary Figures 10-12 are too small to see. It would be beneficial to make the fonts and images larger.

Comment #15

It was difficult for the reviewer to interpret Supplementary Figure 7, especially the reason INSIHGT showed decreasing curve in Supplementary Figure 7B in contrast to Figure 1C. Additional explanations in its figure legends and main text on the page 6 would be helpful.

(Remarks on code availability)

The reviewer was unable to evaluate the code owing to the limited knowledge.

Version 1:

Reviewer comments:

Reviewer #1

(Remarks to the Author)

The authors have made significant and well-justified revisions to the manuscript, addressing the concerns comprehensively. I appreciate the detailed explanations provided and the considerable effort made to address each point. The clarification of terminology in both the text and figure captions, along with the addition of schematics to explain the bulk/cut staining, greatly enhances the accessibility of the manuscript for non-experts. I also appreciate the authors for conducting the imaging experiment on the archived sample expressing tdTomato in place of the suggested positive control, effectively demonstrating the labeling efficiency of INSIGHT. I therefore recommend the manuscript for publication in its current form.

(Remarks on code availability)

The code has improved substantially, with added comments and documentation making it easier to implement and run smoothly.

Reviewer #2

(Remarks to the Author)

I'm very happy with the authors' review, which I found excellent. I have no further comments.

(Remarks on code availability)

Reviewer #3

(Remarks to the Author)

The authors addressed some major concerns; however, some points are still to be clarify:

- Point 3: The text in supplementary figure 11-12-13 is still not visible. Please split the images in more figures to make it possible to read the text above the images.
- Point 4. In the supplementary table please provide specific information about the objectives that have been used, not only the magnification of the objective (e.g Leica 10x ApoPlan...)
- Point 6: I think that the explanation you provide about the restriction of human brain imaging with Mesospim should be added to the text in order to give this information to the reader (as well as the rebuttal figure you presented). Light sheet microscopy has indeed various constraint but is a widespread technology, knowing its limits can help other scientist to know where and how to develop new solution to make it more efficient in human brain imaging.
- Point. 7. As point out by the authors in the images is difficult to discern any features. To provide qualitative information about the accuracy of the segmentation the author should provide from each magnify insets the segmentation of the cells obtained with CellPose. Scale bars info is missing in supp fig 19 and 20 legend. Please add in the text that the multi-round is performed on mouse tissue, since the previous paragraph is referred to human brain tissue clearing a reader can think that the protocol is applied on human tissue.
- Point 8. The accuracy of the segmentation is referred to which marker? In order to finely characterize the analysis you should provide the dice coefficient for all the markers.
- Staining and clearing tissue from various organ and different species usually require specific adjustment of the protocols. Please add precise information about the incubation time needed for the clearing and staining of samples of different organs and from the different species and not only for the different volume dimensions as presented now in the supplementary information. Moreover, please give information about the applicability of the multi round labelling to different species (in you have tested it, or if not explicit it).
- In table 1 please add a column where you specified against which organ/tissue you have test INSIHGT and the species (indeed, human brain tissue staining is more difficult compared to mouse, having this information will help the user to not test all over again the antibodies).
- Concerning the preparation of human brain tissues, from the method section, it is unclear how long the tissue was fixed in PFA after the perfusion before performing the clearing, and if some additional steps were involved (e.g, washing and/or preservation steps), please clarify. Moreover is not specify the age of the donor, this is very important to estimate the myelin content inside the tissue, if possible specify it or provide an approximation.

(Remarks on code availability)

Reviewer #4

(Remarks to the Author)

(Remarks on code availability)

The code has been extensively revised, commented, works perfectly and does what is documented. Perfect.

Data for alternative methods are available and the claimed effect is evident in the data presented. The data used for the iDISCO method appear to be of lower quality, but this does not invalidate the results.

The robustness of the metrics as a function of cell detection sensitivity is validated.

Thank you for your work.

Reviewer #5

(Remarks to the Author)

(Remarks on code availability)

Reviewer #6

(Remarks to the Author)

The authors have satisfactorily addressed the concerns raised by the reviewer.

(Remarks on code availability)

Point-by-point Response to the Reviewers' Comments

Overview of revision

We thank all the reviewers for the very constructive and thoughtful feedback, which substantially improved the quality of our manuscript. The reviewers' comments have been

- copied in black,
- our responses in blue,
- and whenever appropriate the references to the current main manuscript and supplementary file highlighted in yellow.

The benchmarking code has undergone substantial clean-up and revision to make the analysis more robust, including the correction of two major mistakes - the resizing that accidentally mixed the bulk- and cut-staining channels, and the incorrect scale of voxels that underestimated the penetration depths. These have been rectified and a reanalysis showed no change to the conclusions.

We have also added an illustration to demonstrate to make the interpretation of the benchmarking results easier and clearer.

Additional information and discussion, wherever missing or required, have also been added.

Writings have been made more scientifically solid.

The authors thus sincerely ask the reviewers to help with the review process once again.

Response to Reviewer 1

Reviewer #1 (Remarks to the Author):

In this manuscript, the authors presented an elegant and practical concept to achieve homogenous deep 3D histochemistry termed INSIGHT. The authors achieved increased tissue penetration and demonstrated the labeling of macromolecules in tissues down to centimeter depth using the weakly coordinating superchaotropes, and reinstated the antibody-antigen binding using macrocyclic compounds for in situ host-guest reaction. The authors benchmarked their technique with existing deep 3D immunohistochemistry approaches using a rigorous approach, e.g., bulk-staining vs. cut-staining, showcased the superior performance of INSIGHT, and evaluated an impressive list of over 300 antibodies for INSIGHT. The authors demonstrated the versatility of their approach to different biological targets, including small molecules, proteins, lectins, and nucleic acids, various tissues, volumes, and origins. They further showed how the method can be adapted to clinical samples without apparent tissue destruction. The creative application of a simple concept to achieve fast and deep histochemistry is impressive. Overall, the innovative study is well-designed and technically sound, with carefully designed control experiments. The claim is supported by the data, and the protocol could benefit the broad community of 3D histopathology and spatial biology. As such, the manuscript is suitable for publication in Nature Communications after addressing the following comments.

Major comments:

1) The concept of using weakly coordinating superchaotropes to modulate the antibody solubility and tissue penetration depth is interesting. How was the working $\text{Na}_2[\text{B}_{12}\text{H}_{12}]$ concentration obtained? What is the reason for the different concentrations used during screening and INSIGHT protocol (0.1 M vs. 0.25 M)? Have the authors systematically evaluated how the concentration affects the superchaotropic effect on the targets? Would increasing or decreasing concentrations increase/decrease the penetration depth?

- Thank you for the question. Our initial testing of 0.1M for various superchaotropes is because some of the salts (e.g., phosphotungstates) are not very soluble, which at times precipitates with prolonged incubation with phosphate buffer.
- Once we narrowed down to $\text{Na}_2[\text{B}_{12}\text{H}_{12}]$ as the ideal candidate, we optimized the concentration against penetration depth to 0.25M. Lower concentrations of $\text{Na}_2[\text{B}_{12}\text{H}_{12}]$ predictably decrease penetration depth and uniformity.
- Increasing the concentration beyond 0.25M has small effects in increasing penetration depth, but also at the expense of increased cost. For extremely large tissues for dense antigens, simply prolonging the

staining time leads to better penetration depth as a better alternative. This has been supplemented in the protocol section (in supp materials page 6, the last point on the section “Notes”).

2) In the manuscript, the authors mentioned that IgG needs two-step staining, as in primary and secondary staining. In contrast, Fab can be done in a one-step manner by co-incubating the primary and secondary antibodies with the sample. What was the primary reason? Additionally, it would help if the authors could add a discussion on F(ab)₂. Can it be used in single-step staining or two-step staining is required?

- This is due to the monovalency of the Fab fragments, which allows one-step incubation of the primary antibodies (usually in the form of IgGs) without causing clumping. Despite Na₂[B₁₂H₁₂] can help solubilize and reduce protein-protein interactions, the inhibition is not infinite. Especially for F(ab)₂ and IgGs, the doubled binding sites, shown to complement each Fab domain's orientations, effectively doubled the reaction rate¹ which can lead to primary antibody crosslinking and clumping. The nearly doubled and tripled sizes of F(ab)₂ and IgGs, respectively, also lead to a halved and one-third reduction in mobility, respectively, by the Stokes-Einstein's relation, which may lead to their enrichment at the surfaces. The overall effect is more intravascular precipitates and non-uniform labelling, which is what we observed experimentally. This is similarly observed when we use fluorescently tagged streptavidin (usually quadrivalent) with biotin-conjugated secondary Fab fragments, which rapidly leads to clumping and even shallower penetrations.
- We have added the discussion on F(ab)₂ on supp materials page 5 as notes on the procedure.

3) It would be helpful to have a positive control to evaluate the labeling efficiency of INSIGHT. This can be achieved by using a protein target that expresses a GFP fusion protein and assessing anti-GFP immunostaining. For instance, using antibodies such as Abcam ab252881 or ab6673 (refer to Table S2), evaluate the number of cells exhibiting both GFP and Alexa Fluor 647 signals relative to the total number of GFP+ cells. Or at least some extended comments would be helpful, e.g., under the context where the tissue target may be low-abundant.

- We thank the reviewer for the constructive advice. We did not perform such an experiment because we lacked a mouse model in our laboratory expressing a GFP-fusion protein or other fluorescent proteins, and the use of methanol-based dehydration (which is most efficient in our hands) quenches endogenous fluorescence anyway, preventing direct GFP fluorescence and anti-GFP penetration comparisons.
- We understand the concern about incomplete labelling due to the principles of INSIGHT on first globally inhibiting antibody-antigen interactions and then reinstating them deep within the tissue, and concerns may arise that the reinstatement may not be efficient deep within the tissue. Since we also do not have live animals expressing endogenous fluorescence in our hands, we retrieved an archived sample (fixed in formaldehyde for ~3 years) with GCG promoter-driven Cre expression and AAV containing FLOX-ed tdTomato in the brainstem region. The result is a sparse expression of tdTomato+ve fibres in the brain. We found despite heavy fixation, the deep, sparsely labelled fibres can still be visualised near the thalamus (**Rebuttal Fig 1**). This suggests the efficiency of the in situ host-guest reaction is equally efficient at depths and hence permits visualization for deep sparse targets. The high non-specific staining signal (dot and blob-like) is related to over-fixation. It is not observed with other samples throughout the manuscript or when using this antibody (Abcam A121690).

Rebuttal Figure 1. Whole brain labelling and visualization of GCG-dependent tdTomato-positive fibres (using GCG-Cre line mouse with AAV-flox-tdTomato injection into the brainstem). INSIHGT labelling was performed using Goat anti-tdTomato and Donkey anti-Goat Fab fragments (AlexaFluor-594 labelled). The left image shows the overview maximum intensity projection. The right image is an enlarged z-slice in the yellow-boxed region near the thalamus-hypothalamus interface. Yellow arrowheads point to some of the fibres.

Minor comments:

1) The terms weakly coordinating ions, weakly coordinating anions, and weakly coordinating Superchaotropes are interchangeably used and could be clearer. For instance, “We first pre-screened the weakly coordinating anions by immunostaining for parvalbumin in 1mm³ of mouse cortex tissue cubes in the presence of 0.1M weakly coordinating superchaotropes, after 1 day of incubation at room temperature the staining solution was aspirated and 0.1M corresponding cyclodextrin was added and incubated overnight.” It may be useful to have a sentence defining the connection between these terms.

- Thank you for the comment. The term “weakly coordinating superchaotropes” refers to a subgroup of weakly coordinating anions that we hypothesized to inhibit protein-protein interactions. These have been clarified in the manuscript when we used it with the aim of inhibiting protein-protein interactions non-denaturatively (main manuscript page 5).

2) In the section "In situ host-guest chemistry for three-dimensional histology," the author mentioned, "To quantitatively compare the signal, we segmented the labeled cells and compared the ratio between the deep immunolabeling signal and the reference signal against their penetration depths." It would be clearer for the readers if the terms "bulk staining" (deep immunolabeling) and "cut staining" (reference signal) were defined in the text or methods.

- Thank you for the comment. The bulk- and cut-staining method was based on a previously published article². (main manuscript page 6) and illustrated the experiment principles in **Supp Fig. 4a** (supp materials page 11).

3) In the same section, the authors wrote, “This is evident as the cut-staining intensity profile of INSIHGT was similar to that of iDISCO (Fig. S7) which has identical tissue pre-processing steps.” However, in Fig S7, the profiles were not that similar. Could the authors elaborate on the observation?

- Thank you for highlighting the ambiguity in our statement. We have clarified it (now in main manuscript page 7) to emphasize the relationship between cut-staining intensity and penetration depth, focusing on the role of coincubating antibodies with [B12H12]²⁻ in INSIHGT. When complexed and washed out, this results in different cut-staining intensity profiles as a function of the cut-staining penetration depth, namely, a steep exponential decrease in cut-staining intensity with increasing cut-staining penetration depth, similar to iDISCO. This distinguishes it from heavily permeabilized tissues (e.g., CUBIC-HV, SHANEL) or effective deep penetration in INSIHGT bulk-staining.
- If the reviewer notes differences in signal-to-noise ratios in **Supp Fig. 7a** (supp materials page 14), we think it may be due to variations in bulk-staining performance. At deep bulk-penetration depths, fewer antigens labelled during bulk-staining leave more reactive antigens for cut-staining, hence the apparent brighter image in the iDISCO case in **Supp Fig. 7a** (supp materials page 14). We also acknowledge potential biological variations and small differences in manual cutting, as well as residual [B12H12]²⁻-complexes in the tissue affecting the cut-staining slightly. However, as the scatter plot and regression in **Supp Fig. 7b** (supp materials page 14) were based on segmented soma and normalized cut-staining intensities, we believe the conclusion that [B12H12]²⁻ has minimal permeabilization on tissues remains reasonable. These factors also likely do not impact our conclusions about the homogeneous penetration in INSIHGT which we demonstrated throughout the manuscript.

4) In Supp Fig 1. A, it would help to mention whether the shown images are bulk staining or cut staining channel in the figure caption. From the caption from Figure 2B, should it be cut staining channel? It would also help if the ‘mol’, ‘pur’ and ‘gran’ labels on the S1A were defined in the figure caption. Supp Fig 1. B, is the blue stain DAPI?

- Thank you for the comment. The images in **Supp Fig. 1a** are taken with samples that underwent bulk staining. We have supplemented experimental details in the caption accordingly. (supp materials page 8)

5) Supp Fig. 4 is somewhat difficult to interpret, especially the two columns representing shallow ($\frac{1}{3}$) and deep ($\frac{2}{3}$) imaging z-depths. Could the authors elaborate on it? Additionally, it is unclear what host/guest system is used in the example shown.

- Thank you for the constructive advice. Indeed the illustration could have been clearer. We have modified the colour coding representing bulk- and cut-staining to be consistent with that chosen in a previous publication and illustration in **Supp Fig. 4a²** (supp materials page 11), where bulk- and staining-intensities are now always shown in magenta and green, respectively. Additional annotations have also been provided in the figure.

6) Figure 3A, the z-depth definition for color code appears missing.

- Thank you for the reminder, this has been supplemented accordingly (main manuscript page 35).

7) Alexa Fluor 647 was selected to minimize the interference from tissue autofluorescence. Could the author comment on the use of other NIR dyes, such as Cy5 and ATTO 647N? Alexa Fluor 647 is known to photobleach quickly.

- We have also used Cy5, Atto647N, SeTau-647, Cy5.5, Dylight649, iFluor647 with success.
- To help the readers navigate through the tested dyes, we have now supplemented a table (**Supplementary Table 3**) (supp Excel file).

8) On page 39, the INSIGHT protocol lists the equations for the minimum staining time and post-incubation time. The equations are very useful for someone reproducing the results. Are equations empirically determined? For instance, if a tissue section is 10 mm thick, would the staining time be 72 h? Would these equations be generally applicable to a variety of tissues? Some comments would be helpful.

- The equations are empirically determined and shall be applicable to various tissue types, at times depending on the antigen densities. We provide a clearer discussion on the empirical choice of incubation times given by the equations, the assessment of penetration uniformity, and when to

prolong/ shorten the incubation time on the main manuscript page 14 paragraph 2, and in supp materials pages 5 and 6, last point of “Notes”.

9) It’s exciting that the authors observed membrane filaments in tissue (Fig 7). Elongated membrane fibers, such as tunneling nanotubes, have been reported by in vitro studies for intercellular material transfer in various cell type (PMID: 34795441, 25571977, 37202391). Some discussions of observation with existing studies would help enhance the significance of the observation.

- Thank you for helping to draw these connections and improve our discussions, we have now added the information (main manuscript page 11, 2nd paragraph).

Reviewer #1 (Remarks on code availability):

See the attachment "INSIGHT_code_review_NC.pdf" with screenshots.

- Thank you for noting the code running errors. We have entirely rewritten the code due to another Reviewer’s comments and should now run fine.

Response to Reviewer 2

Reviewer #2 (Remarks to the Author):

In this paper the authors developed a user-friendly 3D histochemistry method In situ Host-Guest Chemistry for Three-dimensional Histology (INSIHGT). This method has several advantages, including homogeneous probe penetration up to centimeter depths, quantitative and highly specific immunostaining signals, a fast and affordable workflow that can accommodate different tissue sizes and shapes, simple immersion-based staining at room temperature that can be scaled and automated, and direct application to unlabeled mouse and human tissues using off-the-shelf antibodies.

In this paper they performed a comparison between INSIHGT and the previous 3D technique such as (SHANEL, IDISCO, SHIELD, CUBIC, eFLASH, mELAST) to show the difference in antibodies and probe penetration in the samples. They evaluated 357 antibodies and discovered that 323 of them (90.5%) produced the desired immunostaining patterns when manually confirmed against the human protein atlas and/or current literature which is a high number of comparisons with the other techniques. They also analyzed INSIHGT-treated samples for RNA integrity number (RIN) and whole genome DNA extraction. They discovered that each phase of the INSIHGT methodology did not result in a substantial decrease in RNA integrity number (RIN). For that reason, this technique is particularly significant for the evaluation of the protein and RNA in the same sample.

Finally, they demonstrate the cost-effectiveness of INSIHGT, a three-dimensional (3D) spatial biology approach based on superchaotropes and host-guest chemistry. This allows for multi-modal readout of tissue biomolecules in biological systems at centimeter scales.

Here are some questions for the authors:

1. What is the magnification for 3D imaging?

- The magnification varies between 1.6x to up to 40x. The displayed scale bars provide a rough estimation of the magnifications of the samples during imaging. We have added **Supp Table 4** (**supp Excel file**) detailing all the basic acquisition parameters for the images displayed.

3. Can you provide the number of days needed for staining the full adult mouse brain for each approach compared to INSIGHT (for the experiments you conducted)? As indicated, the stages for iDisco are similar to those for INSIGHT, thus I'd want to know the duration of the staining step.

- Thank you for the comment. The detailed duration of each staining step of each staining protocol in the benchmarking experiment has now been made clear in the **Methods** section (**main manuscript pages 17-18**). The general case (representative of a whole mouse brain) has been discussed and illustrated in **Fig. 2e** in detail (**main manuscript page 34**).

4. Please provide further information on the type of probe used, where it was ordered, and the time of incubation for detecting mRNA and DNA. The step of mRNA detection is not clear.

- Thank you for the reminder, we have added detailed information on the probes (**Table 1**) (**main manuscript page 21**) and FISH procedure in the methods accordingly (**main manuscript pages 20 and 21**).

5. Are there any control trials for 2D mRNA detection to compare expression levels to INSIGHT?

- We did not compare 2D mRNA detection to compare expression levels as each method has its own sampling bias, namely, the 3D method may result in lowered detected expression levels due to penetration problems, imaging sensitivity, sample damage due to INSIHGT etc, while the 2D method has sampling biases on the exact 2D slice sampled, optical sectioning thickness, and suboptimal preservation of RNA due to thin slice and sectioning with environmental contamination.
- If the primary concern is whether INSIHGT chemistry will destroy RNAs, our previous manuscript version has provided an RIN analysis to demonstrate post-INSIHGT RNA integrity. We have now further supplemented the lncRNA differential gene expression (DEG) analyses by RNA sequencing for

non-INSIHGT versus INSIHGT-treated case, which showed essentially no RNA damage, now in **Supp Fig. 15c** (supp materials page 22). INSIHGT is hence the first method that shows no differentially expressed genes by RNA sequencing after 3D imaging, compared to a control sample. The implications of this have been added in the **main manuscript page 9, first paragraph**.

6. Have you used mRNA detection for entire tissues, such as mouse brains or any large samples?

- We are actively developing high-plex mRNA for entire organs, which will be an upcoming study on its own to avoid defocusing the story of addressing the protein-based probe penetration problem.

Reviewer #2 (Remarks on code availability):

The authors have clearly identified the limitations of current immunolabeling methods for obtaining accurate 3D quantification using light sheet microscopy.

The proposed method is undoubtedly of great interest to the scientific community and I've focused more specifically on the claim for homogenous probe penetration, in which I have a particular interest.

Although I was able to run the scripts that generate the figures from the analysis results provided, I was unable to reproduce these results from the data supplied.

In summary, I was unable to reproduce all the results due to:

1- missing data for alternative methods.

2- the information associated with the data is unclear. Data and associated masks are not provided in the same dimensions - channels are interleaved but not clearly labeled - code is very sparsely commented, which can quickly lead to processing errors.

3- resizing operations in `benchmarking_analyze.m` don't seem right to me and can introduce a significant bias in correlation analyses between the two channels.

4- cell identification and segmentation methods are correct, but extremely parameter-dependent. The slightest change in the input data and/or parameters can therefore dramatically alter the results. These parameters should be clearly indicated at the beginning of the script, along with all other parameters related to the analysis. This is particularly important when you need to compare several alternative methods, to make sure that the results presented depend on the specificities of the methods and not on the parameter settings.

More specific indications of the problems encountered are presented below, using as support the README file provided in code-ocean (see attached pdf).

I am sure that optimized analysis code / data / documentation must be available and I remain at the authors' disposal to test them and make them easily testable by the scientific community.

Thank you for your work.

- We are very grateful to the reviewer for spotting the mistake in `imresize3` and resizing operation - which was mistakenly applied before separating the intercalated channels in the tiff stack. These have been rectified and we would like to respectfully ask for the reviewer's expert re-assessment. To avoid such errors, we have also separately saved the image channels and tissue masks in a single folder, all resized and downsized to the same isotropic dimensions using `ImageJ`.
- To address the parameter dependence of the analysis, we have modified the segmentation code to depend only on the expected neuron diameter, which can be set at the start of the script. We chose 15um by manually measuring the observed diameter of the PV+ve neurons in our images. The post-LoG filtering threshold (set to 0.025) was based on a manual visual inspection of the accuracy of the segmentation. We fixed this threshold for all samples as these were benchmarking samples that were processed and imaged using identical parameters. Finally, we performed a sensitivity analysis and varied the expected neuron diameter from 12 to 18 um (see **Rebuttal Fig. 2** below), which did not affect the conclusions that (1) INSIHGT substantially improves the homogeneity of staining across depth, and (2) after complexing and washing off [B12H12]₂- the penetration depth in the cut-staining (right column, related to **Supp Fig. 7** in **supp materials page 14**) reduced significantly. The penetration enhancement dependent on co-incubating [B1H12]₂- is also evident that INSIHGT utilises much shorter staining durations (1 day in the benchmarking experiment).

Rebuttal Figure 2. Sensitivity analyses on varying the single segmentation parameter - the expected neuron diameter, which is related to Fig. 2C (main manuscript page 34), Supp. Fig. 6b (supp materials page 13) and Supp Fig. 7b (supp materials page 14).

- Based on this new analysis, the figures have been updated for Fig. 2c, 2d (upper panel, we confirmed the analysis for c-Fos to be correct, in main manuscript page 34), Supp Fig. 6b (supp materials page 13), Supp Fig. 7 (supp materials page 14), and Supp Table 1 (supp materials page 37). For Fig. 2d upper panel (main manuscript page 34), the Pearson correlation coefficient is now 0.512 (95% CI: 0.467 - 0.554).
- We also took the opportunity to substantially clean, refine and clarify the script, with better documentation and faster running speed (especially by uploading the isotropic and downscaled images) of the entire benchmarking dataset to the Code Ocean repository.

Response to Reviewer 3

Reviewer #3 (Remarks to the Author):

The study proposes INSIHGT “In situ Host-Guest Chemistry for Three-dimensional Histology” a new method to perform homogeneous probe penetration on both mouse and human tissues, to perform specific and multiple markers immunostaining and RNA staining. The proposed protocol is simple and easy to implement in any lab and compatible with scaling and automated analysis.

The work is innovative and will help the scientific community towards the routinely application of 3D histology. However, I have some important points that need to be clarified in order to demonstrate the replicability of the proposed results.

- It is not clear from either the abstract, introduction, or results with which kind of clearing the INSIHGT approach was optimized, only by looking at the method is possible to understand it. Please add it when presenting the INSIHGT method and its identification.

- Thank you for the suggestion. We have further added this discussion in the Introduction and Results (main manuscript pages 3, 6, and 7, respectively), and added more specifications in the protocol notes section (supp materials pages 5 and 6)

- The authors said “Notably, after washing, only a negligible effect of [B12H12]₂-treatment will remain within the tissue.”: to verify the integrity of the nanostructure of tissues after INSIHGT it is desirable to have some TEM or SEM images that will provide information about the ultrastructure of the tissue after the proposed protocol.

- Thank you for the excellent suggestion. The statement (now in the main manuscript page 7) was meant to exemplify the unusual mechanism that [B12H12]₂- helps in probe penetration - which disfavors antibody-antigen binding and such benefit is completely removed after it is complexed by cyclodextrins in the cut-staining stage, inferring little changes made in the porosity of the tissue. Although we lack ready access to TEM and SEM-related methods currently, at this stage, our data suggests that the effect of [B12H12]₂- on probe penetration is based on solvent perturbation, which should result in little structural changes, and if any, would be difficult to assess by electron microscopy as the tissue processing also requires multiple solvent exchanges in ethanol and propylene oxide etc. The fact that little RNA has leaked out based on the newly added control versus post-INSIHGT bulk RNA sequencing (Supp Fig. 15, supp materials page 22), along with previously presented data involving multi-round probing (Fig. 6, main manuscript page 38) and the retention of micrometre-scale structures (Fig. 7, main manuscript page 39) suggests integrity is largely preserved at the micrometre level. We will be looking forward to checking ultrastructural integrity by the users of this method.

- The text in all the figures is too small and not visible, please increase the size of the text to make them visible.

- Thank you for the suggestions, these have been corrected accordingly with considerations for constraints on the figure sizes.

- Please specify the microscope used for every image presented and the resolution used for the acquisitions so as to permit the correct evaluation of the results. Moreover, please give more details about the microscopes and the configuration used for the analysis. E.g: how many laser lines were present in each microscope, which objectives were used, etc.. All the information needed to repeat the experiment by other groups need to be provided. For example: for the 7 channels acquisition with confocal microscopy of 2mm depth samples there is no information about the lasers used, the objective, and therefore the resolution of the images.

- Thank you for the reminder, these have been supplemented in Supplementary Table 4 (supp Excel file) and the Methods section (main manuscript page 20). The maximal number of channels acquired simultaneously was 6 (in Fig. 4d, main manuscript page 36; and Fig. 4h, main manuscript page 36), with the next as 5 channels (in Fig. 6 main manuscript page 38 and Supp Fig. 17 supp materials page

24), which is dependent on the use of the fluorescent dyes, lasers and tunable emission filter settings to minimize spectral overlap using the default suggestion from the Leica LAS X acquisition software or planning using online spectraviewers.

- Please add some z-stack reconstruction to demonstrate the good penetration and contrast of the staining in the samples, 3D reconstructions are very nice to look at, but does not permit to evaluate correctly the staining capability.

- Thank you for the suggestion. An entire scroll-through view of whole mouse brain calcium-binding protein staining (Supp Movie 1 for Fig. 3j-l in main manuscript page 35) and 3.2mm-thick human cerebellum (Supp Movie 3 for Fig. 8a-f in main manuscript page 40) was provided, as for a z-stack reconstruction view for the 3 mm-thick cerebellum slice (Supp Fig. 26 in supp materials page 33). Fig. 3e is the near-mid-transverse slice of the whole mouse brain, which should have shown the thickest dimension of the tissue. We have added the approximate Z-depth of imaging in the figure (which was oblique and hence the approximate depth only) to help the readers understand the depth of imaging for the staining. Further z-stack reconstructions have now been supplemented for the dense kidney labelling (Supp Fig. 9a,b, in main manuscript page 16) related to Fig. 3c-d in main manuscript page 35) and the aged whole mouse brain staining (Supp Fig. 9c in supp materials page 16, Fig. 3e-i in main manuscript page 35).

- In figure 5 to demonstrate the penetration of the antibody inside the tissue the sample was cut and image with confocal microscopy, why the authors did not provide also a 3D reconstruction of the whole block with the mesospim? Even if the resolution is lower, it can detect the signal from the inner part of the sample proving the good antibody penetration. Then to visualize the sample at higher resolution it can be cut and image with confocal microscopy. I suggest the authors analyze an additional human brain sample with the dimension of ~2 x 2 x 3 cm³ (or the larger portion INSIHGT can analyze) and image it with the mesospim to prove the good penetration of the antibody deep inside the sample.

- Thank you for the suggestion. The large specimens prohibited good tissue clearing, and hence the subsequent fair assessment of homogeneous penetration. This is especially problematic with light-sheet microscopy where the excitation and emission paths were decoupled. The resulting image became blurry and difficult to ensure specificity was achieved deep within the sample. We attempted mesoSPIM imaging for our largest demonstrated human sample (Fig 5d in main manuscript page 37, see attached Rebuttal Figure 3 on the GSL-I lectin channel to illustrate the problem) but did not include it in the manuscript to avoid confusion. Nonetheless, the left side illumination at Z-depth = 5,784um (upper panel, mid-row) still shows vaguely discernible vessels, which once cut has been verified in Fig 5d, f in main manuscript page 37. Another centimeter-scale, completely tomographically imaged sample was provided for the MRI-imaged sample for phosphorylated aSyn in Fig 5g-i (stated in Supp Table 4 in supp Excel file).

Rebuttal Figure 3. Original MesoSPIM imaging attempt of the sample in Figure 5d-f in main manuscript page 37. The sample did not clear well, thus prohibiting an adequate evaluation of staining quality near the middle of the sample. Severe scattering and attenuation of excitation light led to differences in images by left or right-sided illuminations.

- For the 7 imaging rounds of 28 neuronal marker in 2 mm-thick hypothalamus slice, please provide images of all the separate channels (e.g for one plane of the sample) in order to provide to the reader a clearer picture of the data obtained with the analysis and the quality of the staining since merged images are very difficult to interpret. Please provide information of the resolution used for the acquisition and the time needed for the acquisition and the analysis in order to give information about the possibility of using the approach and the possible scaling up for routinely analysis.

- Thank you for the suggestion. We have added all rounds of the images for the readers' reference in Supp Fig 19-20 in supp materials pages 26 and 27. The resolution and acquisition parameters have been provided in Supp Table 4 in supp Excel file. Note that for some markers the expression levels are very homogeneous at such a magnification (using a 10x objective) and appear contrast-free (e.g., GABRG2, galanin), making it difficult to discern any features. We have nonetheless verified the

expression patterns to be consistent with the Human Protein Atlas and the literature, whenever available.

- For the multiround analysis please provide the accuracy of cell segmentation obtained using Cellpose 2.0 and the quantity of GT that was performed for each analyzed channel in order to use the tool (or if the author used a pre-trained dataset). Moreover: cellpose is a software that was presented as a method to analyze image in 2D. Since the reconstruction was performed in 3D how the authors used the software to obtain 3D analysis of the samples? Why they didn't used alternative methods developed for 3D analysis as StarDist or BCFind?

- Thank you for the suggestion, we have now computed the accuracy of segmentation using the following parameters:
 - Dice Coefficient: 0.9354 ± 0.0596
 - Jaccard Index: 0.8824 ± 0.1023
- for our custom-trained model using the Cellpose software. This has been supplemented in the methods (main manuscript page 24).
- For the segmentation, the ROIs were trained based on xy-slices and xz-slices on summed channels across all rounds for maximal capturing of cell morphology. We then used the `-do_3D` option in Cellpose to implement the 3D segmentation, which sequentially segments the image stack in xy, xz, and yz and fuses the resulting masks into 3D cell masks.
- We thank the reviewer for suggesting the use of alternative methods such as StarDist and BCFind - which seem excellent alternatives for processing these datasets and we shall explore them in our future studies. We weren't aware of the techniques and hence adhered to Cellpose as per our previous practice. We believe they are all satisfactory given our image qualities as a pilot proof-of-concept demonstration and have provided this information for the readers' consideration in the main manuscript page 10, the paragraph "Volumetric spatial morpho-proteomic cartography for cell type identification and neuropeptide proximity analysis".

- The behavior of the DBSCAN clustering algorithm is influenced by two internal parameters: the search radius ϵ around separate data elements and the minimum number of data points within such a radius required for an element to be considered as belonging to a local "high-density" cluster. Together, these parameters define a minimum threshold on the local data density and thus affect the density and number of the resulting clusters. What are the values adopted by the authors, and what criteria drove the selection of such values? Did the authors evaluate the possible different outcomes of different (ϵ , Nmin) combinations?

- Thank you for the suggestion. We implemented DBSCAN's default choice of $\epsilon = 0.6$ in the UMAP 4.4 Matlab package with settings to "cluster detail" set to "very high" clustering, which gives visually good clustering results and re-embedding in 3D biological space by eye. We have supplemented these crucial analysis parameters in the Methods section (main manuscript page 24). For the choice of 'min_dist' and 'n_neighbors', we referred to this guide for recommended default choices³. The choice of 'min_dist' = 0.05 rather than 0.1 was because it gave more tightly packed clusters, hence the subsequent DBSCAN clustering would be less sensitive to the choice of ϵ .
- We did not further evaluate the possible different outcomes of parameter combinations as our study was more focused on using novel chemistry to affordably deliver 3D multiplexed, near-ground-truth data across depth for subsequent spatial bioinformatics, which is out of our team's expertise. The demonstration of one sample 3D hi-plex IHC was therefore an illustrated example only to demonstrate the feasibility of the chemistry and workflow advantage (fast 3D deep multiplexed immunostaining)

- In figure 6 for both the cell and NYP fiber segmentation please provide high resolution images of the segmentation obtained with the analysis. From the small images presented in figure 6 the quality of the cell segmentation is not appreciable, and the NYP segmentation is not provided at all. Please include information about the accuracy of the segmentation and the presence of false positive and negative recognitions.

- Thank you for pointing out the vague fiber representation, which is partially constrained by figure sizes. Our intensity-based segmentation performance on an illustrated slice has now been provided in **Rebuttal Fig 4**. Figures demonstrating the quality of NPY staining have been provided in **Rebuttal**

Fig 4, along with the segmentation based on its intensity. For cells, the segmentation was by Cellpose and is a crude representation of the cell borders encompassing the immunostaining signals - a workaround due to the lack of markers or imaging methods for visualizing the exact border for the fractal-like morphologies of the cells in the brain.

- Since the segmentation for NPY+ve fibers was simply intensity-based, hence we did not compute the Dice coefficient but rather relied on manual inspection and cross-comparing with the literature results only. The structures are also too small (near point-like) for accurate and easy manual drawing of the masks.

Rebuttal Figure 4. A non-linearly registered NPY staining image and intensity-based segmentation results related to **Figure 6** in main manuscript page 38. The original image signal is displayed in yellow while the segmentation is displayed in cyan.

- We thank the reviewer for contributing to improving the accuracy and comprehensiveness of the manuscript. We believe as segmentation techniques constantly evolve, the data analysis will further improve. Our point here is that INSIHGT as a novel chemical approach to solving the probe penetration problem can provide sufficiently high-quality images without any prior image processing (e.g., selective removal of nonspecifics, gamma corrections, intensity transforms with custom masks) where state-of-the-art segmentation techniques can be readily applied (constrained by each algorithms' limitations), to the extent that manual providence of ground-truth mask is unambiguous and confident for biologists lacking expertise in image analysis.

- In the discussion the authors say: "Immunostaining penetration homogeneities for larger tissues and denser antigens can be further enhanced": to be more precise please provide the exact depth limit achievable with INSIHGT so that the reader can have a clear knowledge of the possibility of the method.

- Thank you for the suggestion, the additional discussion on the interplay between tissue size and measures on the limits of INSIHGT has now been provided in main manuscript page 14.

Reviewer #3 (Remarks on code availability):

I don't have the expertise to review the code.

Reviewer #4 (Remarks to the Author):

Reviewer #4 (Remarks on code availability):

Reviewer #5 (Remarks to the Author):

Reviewer #5 (Remarks on code availability):

Response to Reviewer 6

Reviewer #6 (Remarks to the Author):

In this manuscript, the authors developed a two-step staining method, named INSIHGT, that enabled multi-modal analyses by simultaneously labeling a wide range of biomolecules in a three-dimensional tissue. The method reported in this well-written manuscript is a breakthrough in many biological fields. I hope that my comments below are helpful in improving the manuscript.

Comment #1

In the Abstract section, introducing INSIHGT in a more concrete way might be reader friendly. For example, adding “two-step staining (method)” and “simultaneously (visualized)” in the third sentence and “deep inside the tissue (up to centimeter scale)” in the fourth sentence might be informative.

- Thank you for the great suggestion. We have made the suggested writing changes in the **abstract**. We respectfully preferred to omit to state INSIHGT as a “two-step” method as this might confuse the audience on an alternative interpretation of separately incubating the tissue with primary and secondary antibodies.

Comment #2

According to the Microscopy subsection in the Methods section on the page 35, three types of microscopies (confocal, light-sheet and two-photon) were used. For clarity, please consider adding information on the type of microscopy used for image acquisition in each figure legends throughout the manuscript.

- Thank you for the comment, the acquisition parameters and microscopy types have been supplemented in **Supplementary Table 4 (supp Excel file)**.

Comment #3

In the Figure 2C, the fluorescent “ratio” between bulk- and cut-stain was compared. Describing the fluorescence intensity in arbitrary unit, in addition to the ratio, for each staining method would enhance understanding of the experiment.

- Thank you for the suggestion. The reason for the choice of a ratio (as suggested in a previous publication) is that the cut-staining channel was supposed to reflect the ground truth intensities, where each channel’s original fluorescence intensity can be affected by factors other than penetration, notably the underlying biological expression levels, the excitation-emission characteristics of the fluorescent dyes, and accessibility of the antigens (due to steric factors and local environment). The ratio between a deep staining method and true antigen expression level - to the best of our knowledge and design - is the non-biased metric for assessing penetration homogeneity.
- We have added a separate illustration and discussion to better explain the benchmarking experimental design (**Supp Fig 4a, supp materials page 11**).

Comment #4

In the Figure 4B, not in the Figure 4C, the degree of lysine methylations was examined. Please correctly refer to this point in the main text (the second “Fig. 4B-C” should be corrected to “Fig. 4B”).

- Thank you for noting the mistake, we have edited the writings in the revised manuscript (**main manuscript page 8**).

Comment #5

Application of INSIHGT to nucleic acid probes (Figures 4E-G) is one of the key features. In the “Graphical summary of INSIHGT protocol” on the page 42, use of INSIHGT C solution was nicely summarized. Adding brief explanation sentences on the experimental procedure in the corresponding paragraph on the page 8 might be informative.

- Thank you for the suggestion, we have added the information accordingly in **main manuscript page 8**.

Comment #6

The present reviewer was enabled to find the pathological contribution of the subcerulean nuclei in the reference #25 while, according to Figure 5G and 5H, the sampled area is the locus ceruleus (LC). Please check the sampled area and add figure legends for Figures 5G-5I that are missing in the current manuscript. In addition, as is already provided in the Supplementary Figure 15, the full spellings of the brain areas were required in the figure legends of the main text: inferior colliculus (IC), red nucleus (RN), substantia nigra (SN), superior colliculus (SC).

- Thank you for noting we accidentally missed out on the figure legends for Fig 5g-i in the main manuscript page 37, for which we have now added the information in the revised manuscript. We refrained from annotating the subcerulean nucleus because its boundary and location are still somewhat debatable. As a workaround, we have renamed the “LC” annotation as locus ceruleus complex (main manuscript page 9 and in the figure caption, main manuscript page 37) to both address the existing literature naming and the pathobiological significance of the subcerulean nucleus’ role in RBD.

Comment #7

The sentence (“Such a ratio-histopathology approach ... for predicting underlying neurodegeneration.”) might be an overspeculation on the application of INSIHGT. One of the breakthroughs of INSIHGT lies in bridging the microscopic and mesoscopic imaging modalities at a molecular level. Given the latter point, deleting the former sentence from the manuscript would enhance the scientific soundness without reducing the impact of INSIHGT.

- Thank you for the suggestion. We have adjusted our writing accordingly to match the key breakthroughs of INSIHGT as identified by the reviewer (in the main manuscript page 9 1st paragraph).

Comment #8

Twenty-five molecular markers instead of 50 markers were examined in Supplementary Figure 18. Please revise the first sentence on the page 10 (“a subset of 84,139 cells based on 50 markers”).

- Thank you for the comment. We have clarified the meaning of 50 markers in our clustering analysis (25 molecular markers’ mean intensity and their voxel-wise intracellular standard deviations) in main manuscript page 10.

Comment #9

Attempts to link cell morphology and single-cell transcriptomics have been made, for example in a paper (PMID: 35347329). Therefore, the expression on the page 10 (“which is inaccessible to spatial transcriptomics or single-cell multi-omics”) is too strong. Please consider reducing the tone here. Alternatively, the strength of INSIHGT is enabling mapping “proteins” of three-dimensional tissues at a subcellular level, which could be emphasized. Indeed, measuring the distance between NPY fibers and the nearest cell membrane is meaningful because INSIHGT was applied to the three-dimensional tissue.

- Thank you for the comment. We originally intended to discuss 3D cell morphologies that were difficult to infer from spatial transcriptomics and single-cell multi-omics (requiring dissociation of cells) datasets, which is also problematic for 2D cases⁴. However, as technology improves this may no longer be true. We have revised the discussion in the main manuscript page 10.
- The paper provided by the reviewer is an algorithm development work, which complements the data integration problem arising from the datasets generated by wet lab methods such as INSIHGT - as it is an open question on how to merge the information from distinct domains (e.g., cell geometry, molecular expressions, spatial location etc) in the definition of cell types. We look forward to the community to develop such tools for their use.

Comment #10

In an animal model of crescentic glomerulonephritis, the link between the glomerulus tuft and Bowman’s capsule is reported (PMID: 11562404). This linking structure in the normal condition, however, is hardly documented to the best of the reviewer’s knowledge. Please consider reducing the tone in the sentence

referencing # 27-31. Alternatively, the strength of INSIHGT here is that it carries the information on the constituent such as wheat germ agglutinin (WGA) of this linking structure. In addition, adding discussion on the reason INSIHGT enabled visualization of this structure might be informative.

- Thank you for helping to find a potentially related structure - which might have evolved from the filaments in conditions displaying cresenteric glomerulonephritis. We have now incorporated these new findings, and a toned down discussion with a better discussion on the complementarity of INSIHGT in enabling such findings, in the main manuscript page 11.

Comment #11

The reviewer has two minor suggestions in the kidney subsection on the page 10. The inverse relationship reported in Figure 7G is difficult to understand at a glance that would be supported by Pearson coefficient. In addition, the word “overarching (PECs)” is confusing because the linking structure belongs to the podocyte. Please consider these points.

- Thank you for the suggestion. For the inverse relationship, it exemplifies the decreasing probability of finding another filament of similar length along the surface of the glomeruli, while long filaments don't tend to occur in clusters. A plot based on 1/geodesic distance against path length has been considered along with Pearson's coefficient fitting for a $y = 1/x$ relationship but we eventually opted for the presented figure for a more straightforward representation.
- We have corrected the term “overarching” as the “nearest PEC across the Bowmann space” (main manuscript page 11).

Comment #12

The human brain tissue derived from a single DLB patient according to page 31. Therefore, discussion on the pathology of the NF-H-intense inclusions, if applicable, might be informative.

- We have added additional discussions on the potential relationship with DLB on the NF-H intense inclusions (main manuscript page 12), nonetheless, we have also observed them in the normal brain (now added in **Supp Fig 28C**, supp materials page 35), which requires further studies to understand its biological significance.

Comment #13

Adding the reference numbers of approval for the use of human- and animal-derived tissue on pages 30 and 31 are encouraged.

- Thank you for the reminder, we have added the information in the main manuscript pages 15 and 16 accordingly.

Comment #14

Supplementary Figures 10-12 are too small to see. It would be beneficial to make the fonts and images larger.

- Thank you for the suggestion, we have tried our best to maximize the figure text and image display accordingly under the page constraints.

Comment #15

It was difficult for the reviewer to interpret Supplementary Figure 7, especially the reason INSIHGT showed decreasing curve in Supplementary Figure 7B in contrast to Figure 1C. Additional explanations in its figure legends and main text on the page 6 would be helpful.

- Thank you for expressing the difficulty in interpreting **Supp Fig 7** (now in supp materials page 14), which is meant to show that once [B12H12]₂⁻ was complexed by 2HPyCD, the deeply penetrating effects of [B12H12]₂⁻ was completely abolished, and hence a sharp contrast in penetration intensity profile in **Fig 2c** (now main manuscript page 34). We have now added an additional explanation (with illustration in **Supp Fig 4a**, supp materials page 11) on the interpretation accordingly.

Reviewer #6 (Remarks on code availability):

The reviewer was unable to evaluate the code owing to the limited knowledge.

References

1. Galanti, M., Fanelli, D. & Piazza, F. Conformation-controlled binding kinetics of antibodies. *Sci. Rep.* **6**, 18976 (2016).
2. Lai, H. M. *et al.* Antibody stabilization for thermally accelerated deep immunostaining. *Nat. Methods* **19**, 1137–1146 (2022).
3. UMAP API Guide. <https://umap-learn.readthedocs.io/en/latest/api.html>.
4. Dayao, M. T., Brusko, M., Wasserfall, C. & Bar-Joseph, Z. Membrane marker selection for segmenting single cell spatial proteomics data. *Nat. Commun.* **13**, 1999 (2022).

Point-by-point Response to the Reviewers' Comments

Overview of revision

We thank all the reviewers for their effort in re-assessing and helping check our work in detail, which helps in validating the robustness of our technology and the clarity of delivery to the readers.

The reviewers' comments have been

- copied in black,
- our responses in blue,
- and whenever appropriate the references to the current main manuscript and supplementary file highlighted in yellow.

Reviewer #1 (Remarks to the Author):

The authors have made significant and well-justified revisions to the manuscript, addressing the concerns comprehensively. I appreciate the detailed explanations provided and the considerable effort made to address each point. The clarification of terminology in both the text and figure captions, along with the addition of schematics to explain the bulk/cut staining, greatly enhances the accessibility of the manuscript for non-experts. I also appreciate the authors for conducting the imaging experiment on the archived sample expressing tdTomato in place of the suggested positive control, effectively demonstrating the labeling efficiency of INSIGHT. I therefore recommend the manuscript for publication in its current form.

Reviewer #1 (Remarks on code availability):

The code has improved substantially, with added comments and documentation making it easier to implement and run smoothly.

We sincerely thank the reviewer's encouragement and constructive feedback once again.

Reviewer #2 (Remarks to the Author):

I'm very happy with the authors' review, which I found excellent. I have no further comments.

We sincerely thank the reviewer's encouragement and constructive feedback once again.

Reviewer #3 (Remarks to the Author):

The authors addressed some major concerns; however, some points are still to be clarify:

- Point 3: The text in supplementary figure 11-12-13 is still not visible. Please split the images in more figures to make it possible to read the text above the images.

Thank you for the suggestion, these have now been splitted into 5 supplementary figures (Supplementary Text Page 18-22).

- Point 4. In the supplementary table please provide specific information about the objectives that have been used, not only the magnification of the objective (e.g Leica 10x ApoPlan...)

The information was provided in the table description to avoid an overly complicated table, as they all used the same objective for a given magnification. We think this might be missed because of uploading as excel files, the table description has now been provided in the Supplementary text file (page 51-52), and also included in the methods section.

- Point 6: I think that the explanation you provide about the restriction of human brain imaging with Mesospim should be added to the text in order to give this information to the reader (as well as the rebuttal figure you presented). Light sheet microscopy has indeed various constraint but is a widespread technology, knowing its limits can help other scientist to know where and how to develop new solution to make it more efficient in human brain imaging.

We have added the rebuttal figure as a supplementary figure (Supp Fig. 18, Supp text file page 25) and a brief statement in the main text (Main manuscript page 9). The supplementary figure caption discusses the challenges of penetration assessment for large human tissues using light-sheet microscopy, which we feel is easier for the readers when we reference the phenomena described in the figure.

- Point. 7. As point out by the authors in the images is difficult to discern any features. To provide qualitative information about the accuracy of the segmentation the author should provide from each magnify insets the segmentation of the cells obtained with CellPose. Scale bars info is missing in supp fig 19 and 20 legend. Please add in the text that the multi-round is performed on mouse tissue, since the previous paragraph is referred to human brain tissue clearing a reader can think that the protocol is applied on human tissue. Thank you for the reminder about the scale bars - which we have further added in the corresponding figure captions (Supp Text Page 29-30). We also added the “mouse” tissue in Page 10 of the Main manuscript file. Magnified insets of segmentation for each marker does not apply (as explained below), but a view of the segmented masks and cells had been provided in Fig 6e (Main manuscript file Page 38), which we believe is adequate for a chemical method development study.

- Point 8. The accuracy of the segmentation is referred to which marker? In order to finely characterize the analysis you should provide the dice coefficient for all the markers.

In our previous version of our manuscript, we explained in our Methods section in Page 24 of the Main manuscript file: “Before segmentation, all non-vessel channels underwent background subtraction. They were then summed to capture the full morphology of stained cells, followed by segmentation”. Hence only a single Dice coefficient can be provided. The plus-or-minus range is the SD based on 6 excerpted test images. We have clarified these info in the manuscript (Main manuscript file Page 24).

- Staining and clearing tissue from various organ and different species usually require specific adjustment of the protocols. Please add precise information about the incubation time needed for the clearing and staining of samples of different organs and from the different species and not only for the different volume dimensions as presented now in the supplementary information. Moreover, please give information about the applicability of the multi round labelling to different species (in you have tested it, or if not explicit it). Thank you for bringing this issue out. Our experiments followed the recommended empirical formula of staining duration lower-bound estimation (Supp Text Page 5), which is a distinguishing strength as it allows for a standardized, predictable workflow, essential for clinical applications. We have added this discussion in the Supp Text Page 5. Tissue clearing is not the subject of our study so we did not record the duration of clearing (as it is required to incubate until tissue clarity plateaued). We added the explicit statement that we haven't applied multi-round labelling of INSIGHT in other species, though this is theoretically applicable (Main manuscript file Page 14).

- In table 1 please add a column where you specified against which organ/tissue you have test INSIGHT and the species (indeed, human brain tissue staining is more difficult compared to mouse, having this information will help the user to not test all over again the antibodies).

We assume the reviewer is referring to Supp Table 2 on the list of antibodies. If so, such information has now been supplemented (Supp Text Page 41-49).

- Concerning the preparation of human brain tissues, from the method section, it is unclear how long the tissue was fixed in PFA after the perfusion before performing the clearing, and if some additional steps were involved (e.g, washing and/or preservation steps), please clarify. Moreover is not specify the age of the donor, this is very important to estimate the myelin content inside the tissue, if possible specify it or provide an approximation.

Thank you for the reminder, we have now added the information in the Method section (Main manuscript Page 16). We do not know of a way to reliably estimate myelin contents, but we believe this does not change the conclusion that INSIGHT provides 10-20x deeper penetration with improved uniformity.

Reviewer #4 (Remarks to the Author):

Reviewer #4 (Remarks on code availability):

The code has been extensively revised, commented, works perfectly and does what is documented. Perfect.

Data for alternative methods are available and the claimed effect is evident in the data presented. The data used for the iDISCO method appear to be of lower quality, but this does not invalidate the results.

The robustness of the metrics as a function of cell detection sensitivity is validated.

Thank you for your work.

We sincerely thank the reviewer's validation and efforts in checking the code thoroughly, the provided plots are great in visualizing the data.

Reviewer #5 (Remarks to the Author):

Reviewer #6 (Remarks to the Author):

The authors have satisfactorily addressed the concerns raised by the reviewer.

We sincerely thank the reviewer's encouragement and constructive feedback once again.

Comments about the software:

The codes provided by the authors were tested in MATLAB R2022b. Error messages encountered are listed below.

In the 'benchmarking_analyze.m' error at line 99.

Error using imwrite

Unable to open file "D:\NC_review\Test\cell
segmented_all\INSIGHT_org_image_allcells.tif" for writing.

You might not have write permission.

Error in benchmarking_analyze (line 99)

```
imwrite(allcells(:,1), [parentpath 'cell segmented_all' inname '_allcells.tif',]);
```

What is 'deepdata.mat'?

'Concordance_analysis.m' does not output the scatter plot for Fig 2D as mentioned in the readme file.

Error: Brace indexing is not supported for variables of this type.

Error in Concordance_analysis (line 4)

```
sig = data{1,ii};
```

MATLAB R2022b - academic use

HOME PLOTS APPS Search Documentation

EDITOR PUBLISH VIEW

Current Folder: D:\NC_review\Test\Concordance

Command Window

```
>> benchmarking_post_analyze
ans =
    'D:\NC_review\Test\Code'
```

Warning: Converting X to matrix of double.
> In curvfit_attention/Warning/throw (line 30)
In fit>fit (line 130)
In fit (line 110)
In benchmarking_post_analyze (line 35)
Warning: Converting Y to vector of double.
> In curvfit_attention/Warning/throw (line 30)
In fit (line 110)
In benchmarking_post_analyze (line 35)
Warning: Converting X to matrix of double.
> In curvfit_attention/Warning/throw (line 30)
In fit>fit (line 130)
In fit (line 110)
In benchmarking_post_analyze (line 71)
Warning: Converting Y to vector of double.
> In curvfit_attention/Warning/throw (line 30)
In fit>fit (line 130)
In fit (line 110)
In benchmarking_post_analyze (line 71)
>> Concordance_analysis
Error in Concordance_analysis (line 4)
sig = data(1,ii);
fx>>

Workspace

- A
- ans
- Ao
- B
- celldata
- CI
- color
- count
- data
- deepdata
- fitresult
- ifresults
- ix
- IntensityData
- master
- order
- p
- p1
- p2

Editor

```
1 master = [];
2 for ii = [2 4 5 6 7]
3     sig = data(1,ii);
4
5     for jj = 1:2
6         sig(:,jj) = sig(:,jj) - mean(sig(:,jj));
7         sig(:,jj) = (sig(:,jj))/(max(sig(:,jj))-min(sig(:,jj)));
8         p = prctile(sig(:,jj),[10 90]);
9         %sig(sig(:,jj)> p(2)) = 0;
10    end
11    data(2,ii) = sig;
12    master = [master; sig];
13 end
14
15 master = master(all(master,2),:);
16
17 figure
18 scatter(master(:,1), master(:,2), 15, [0 0.7 0.3], 'filled'); hold on
19 % for ii = [2 4 5 6 7]
20 %     sig = data(2,ii);
21 %     scatter(sig(:,1), sig(:,2), 15, [0 0.7 0.3], 'filled'); hold on
22 % end
23 %xlim([0 3]); ylim([0 3]);
24 xlabel('normalized out-staining intensity'); ylabel('normalized bulk-staining intensity');
25
26 fitresult = fit(master(:,1),master(:,2),'poly1');
27 fitresult2 = fitm(master(:,1), master(:,2));
28 fitresult2
29 p = plot(fitresult, '-'); hold on
30 p.Color = [0.3 0.3 0.3]; p.LineWidth = 4;
31 set(gca, 'tickdir', 'out');
32
33 [R,p,RL,RU] = corrcoeff(master(:,1), master(:,2));
34 disp(['Pearson correlation coefficient ' num2str(R(2))]);
35 disp(['P value ' num2str(p(2))]);
36 disp(['coeff upper bound ' num2str(RU(2))]);
37 disp(['coeff lower bound ' num2str(RL(2))]);
38 %
39 error = zeros(100,1);
40 for ii = 1:100
41     P1 = prctile(master(:,1),ii);
42     P2 = prctile(master(:,2),ii);
43     idx1 = find(master(:,1)<P1 & master(:,2)>P2);
44     idx2 = find(master(:,1)>P1 & master(:,2)<P2);
45     idx = [idx1; idx2];
46     error(ii) = numel(idx);
47 end
48 figure
49 plot([1:100], 100*error/size(master,1));
50 set(gca, 'tickdir', 'out');
```

Révisions INSIGHT 2 :

The code has been extensively revised, commented, works perfectly and does what is documented. Perfect.

Data for alternative methods are available and the claimed effect is evident in the data presented. The data used for the iDISCO method appear to be of lower quality, but this does not invalidate the results.

The robustness of the metrics as a function of cell detection sensitivity is validated.

Thank you for your work.

Révisions INSIGHT 2 :

Previous items :

1- missing data for alternative methods.

OK - data for alternative methods are available. The data used for the iDISCO method appear to be of lower quality, but this does not invalidate the results presented.

2- the information associated with the data is unclear. Data and associated masks are not provided in the same dimensions - channels are interleaved but not clearly labeled - code is very sparsely commented, which can quickly lead to processing errors.

The code has been extensively revised, commented on, is working perfectly and does what it says. Perfect.

3- resizing operations in benchmarking_analyze.m don't seem right to me and can introduce a significant bias in correlation analyses between the two channels.

Corrected

4- cell identification and segmentation methods are correct, but extremely parameter-dependent. The slightest change in the input data and/or parameters can therefore dramatically alter the results. These parameters should be clearly indicated at the beginning of the script, along with all other parameters related to the analysis. This is particularly important when you need to compare several alternative methods, to make sure that the results presented depend on the specificities of the methods and not on the parameter settings.

The variation in metrics of interest as a function of detection sensitivity has been taken into account. The robustness of the metrics has been validated. Thank you for your work.

SWITCH-ELAST - zoi: 20

SWITCH-ELAST - Nb Z: 50

SWITCH-ELAST - zoi: 20

SWITCH-ELAST - zoi: 20

SHANEL - zoi: 20

SHANEL - Nb Z: 59

SHANEL - zoi: 20

SHANEL - zoi: 20

INSIHGT - zoi: 20

INSIHGT - Nb Z: 70

INSIHGT - zoi: 20

INSIHGT - zoi: 20

iDISCO - zoi: 20

iDISCO - Nb Z: 56

iDISCO - zoi: 20

iDISCO - zoi: 20

eFLASH - zoi: 20

eFLASH - Nb Z: 70

eFLASH - zoi: 20

eFLASH - zoi: 20

CUBIC-HV - zoi: 20

CUBIC-HV - Nb Z: 86

CUBIC-HV - zoi: 20

CUBIC-HV - zoi: 20